# Network pharmacology predicts combinational effect of novel herbal pair consist of Ephedrae herba and Coicis semen on adipogenesis in 3T3-L1 cells

**Dong-Woo Lim** [1,2], **Ga-Ram Yu** [1], **Jai-Eun Kim** [3]*, **Won-Hwan Park** [1]*

1 Department of Diagnostic, College of Korean Medicine, Dongguk University, Goyang, Republic of Korea, 2 Institute of Korean Medicine, Dongguk University, Goyang, Republic of Korea, 3 Department of Pathology, College of Korean Medicine, Dongguk University, Goyang, Republic of Korea

* diapwh@dongguk.ac.kr (W-HP); herbqueen@dongguk.ac.kr (J-EK)

**Data Availability Statement:** All relevant data are within the paper and its Supporting Information files.

**Funding:** LDW. (only author that was awarded fund) This work was supported by the Basic

## Abstract

### Background

Herbal combinations are regarded as basic strategy in oriental medicine with various purposes. Ephedrae herba (EH) and Coicis semen (CS) are two herbal medicines used to treat obesity in many herbal prescriptions, yet the effect and significance of this herbal pair have not been evaluated.

### Purpose

This study is to elucidate the effect of a novel herbal pair, EH-CS, on obesity and identify the key synergistic mechanism underlying it.

### Methods

We investigated the network of herbs comprising the anti-obesity herbal prescriptions. Using the tools of network pharmacology, we investigated the compound-target interactions of EH and CS in combination to predict their effects in combination. Five EH-CS samples with different EH to CS ratios were prepared to investigate their efficacies in adipocytes.

### Results

1-mode network analysis of herbs in prescriptions based on literature review revealed the importance of EH-CS in anti-obesity prescriptions. The herbal combination comprised of equivalent weights (1:1) of EH and CS most potently reduced mature adipocyte adiposity, although several markers of adipogenesis and lipid synthesis were more suppressed by pure EH. PTGS2 (COX-2 gene) expression, a common target of EH and CS as deduced by compound-target network analysis, was affected by EH-CS extract treatments. However, EH at high concentration (25 μg/ml) notably increased PTGS2 expression without adversely affecting cell viability. However, EH-CS combination of the same concentration markedly decreased PTGS2 gene expression.

Science Research Program of the Korean National Research Foundation funded by the Ministry of Education (Grant no. 2021R1A6A3A01086718). https://www.nrf.re.kr/index The funders had no role in study design, data collection and analysis, decision to publish, or preparation of the manuscript.

**Competing interests:** The authors have declared that no competing interests exist.

**Abbreviations:** BP, biological process; BSA, bovine serum albumin; CC, cellular component; CS, Coicis semen; DL, drug likeness; DMEM, Dulbecco's Modified Eagle Medium; DPBS, Dulbecco's Phosphate Buffered saline; EH, Ephedrae herba; FBS, fetal bovine serum; GO, gene ontology; KEGG, Kyoto Encyclopedia of Genes and Genomes; MF, molecular function; OB, oral bioavailability; ORO, oil red o; PA, palmitic acid; PPI, protein-protein interaction; TCMSP, The traditional Chinese medicine systems pharmacology database and analysis platform.

## Conclusion

These results show that the compounds in CS and EH act in concert to enhance the pharmacological effect of EH, but control unexpected effects of EH treatment.

## 1. Introduction

Drug combinations are often proposed as novel prospective treatments and pose considerable challenges to drug developers [1]. However, in oriental medicine, the concept of multicomponent prescriptions is universally accepted, and such prescriptions are used as primary treatments [2]. There is a perception of herbal pairs used for certain effects of formulae in oriental medicine, like modulating efficacy, toxicity and bioavailability (or absorption) [3]. Recent progress in analysis of herbal medicine on structural similarities at the systems level have elucidated certain pairs of natural compounds work in combination [4], and many authors have reported on the combinatorial effects of various herbal pairs, such as Radix Sophorae Flavescentis- Fructus Cnidii [5], Chuanxiong Rhizoma-Cyperi Rhizoma [6], and Danshen–Honghua (Salviae miltiorrhizae radix-Flos Carthami) [7]. Such studies are on-going, and researchers continue to identify novel herbal combinations that, for example, improve drug efficacies or reduce side effects.

Ephedrae herba (EH) is the rhizome of the perennial plant, *Ephedra sinica*, which has long been used to treat the common cold, arthralgia, and asthma [8], and is now prescribed by physicians to induce weight loss [9,10]. The pharmaceutical potential of EH on metabolic diseases or syndromes is supported by scientific evidence of increased thermogenesis achieved by sympathetic neuron control [11] and the browning effect on white adipose tissue [12]. Furthermore, EH is frequently used with other herbs [13] to obtain various effects and ameliorate EH treatment-associated unsolicited symptoms [14].

On the other hand, Coicis semen (CS, the seed of *Coix lacryma-jobi*) is also used to treat obesity and diabetes in traditional medicine [15]. CS has been reported to have anti-diabetic [16], anti-oxidative [17], and anti-tumor properties [18], which have been attributed to its phytochemical constituents, such as polyphenols, flavonoids, lignans, and phytosterols [15]. Several anti-obesity herbal prescriptions containing CS continue to be used because of its safety [19] and efficacy [20] profiles.

A few researchers have raised questions about possible interactions between the two herbs (EH-CS) for its co-occurrence in many prescriptions. An analysis of precedent research on clinical herbal prescriptions by Song *et al.* revealed that EH and CS are the herbs most commonly used to treat obesity [21]. We also tentatively suggested the possibility of unidentified synergic effects of these two herbs based on a review of common herbs used in anti-obesity prescriptions in previous *in vivo* study [22]. However, these studies only suggested the possibility and did not provide supporting evidence of their effects. Despite the possibility of pharmacological benefits that the empirical use of the combination may have, there has been no intensive research on it. Network pharmacology, an integrative tool for analyzing pharmacological mechanism of herbal medicine [23], can be used to decipher complex interactions between numerous targets and compounds derived from EH and CS.

Prostaglandin-endoperoxide synthase2 (PTGS2) is inducible enzyme expressed in inflamed conditions leading to biosynthesis of prostaglandins [24]. It has been noted that inhibition of COX activity affects adipocyte differentiation via decreasing inflammatory cytokines [25]. A member of the nuclear hormone receptor coactivator family, nuclear receptor coactivator 2

(NCOA2) controls adipogenesis, lipid metabolism, and fat absorption to maintain metabolic balance [26]. Adrenoreceptor beta 2 (ADRB2) is connected to elevated noradrenaline release brought on by exposure to cold, which activates lipolysis and thermogenesis [27]. Interleukin-6 (IL6) is pro-inflammatory mediator which is suggested as cause of systemic low grade inflammation in obesity [28]. By offering a variety of strategies in various metabolic pathways, those genes can be used as targets for the treatment of obesity.

In this study, network pharmacology analysis was used to predict key targets and decipher the combinational effects of EH-CS herbal pair with their constituents. We evaluated the anti-adipogenic effects of EH-CS herbal combinations in a mature 3T3-L1 adipocyte model. To confirm the synergistic effect of EH-CS combinations, we tested the effects of five EH-CS samples prepared using different EH to CS ratios (EH-CS; 0–100, 25–75, 50–50, 75–25, 100–0, percent w/w). Mechanisms probably responsible for the effects of EH-CS as predicted by network pharmacologic analysis were verified in a palmitate-induced inflammatory preadipocyte model treated with separated concentrations of EH-CS sample. Finally, we tried to find the reasonable explanation to describe the different effects caused by EH-CS combinational samples with the clinical implication.

## 2. Materials and methods

### 2.1 HPLC-DAD-UV analysis

Chromatographic analyses of EH-CS and standard ingredients were performed using a HPLC system (Agilent 1260 infinity HPLC, Agilent, CA, USA) equipped with a binary pump, UV-Diode array detector, degasser, and auto-sampler. Standard compounds for EH-CS were obtained from Sigma (Sigma Aldrich, St. Louis, Missouri, USA). Standard compounds and EH-CS extracts were diluted with pure ethanol to 5 mg/mL and filtered. The components of EH-CS were analyzed using an Agilent Eclipse XDB-C18 chromatographic column (150×4.6 mm, 5 μm pore size) at a flow rate of 2 mL/min, a column temperature of 25˚C, and a detection wavelength of 192 nm. Two mobile phase conditions were used; 1) isocratic (A–water with 0.05% formic acid and B–acetonitrile) of A:B = 50:50 consistently for 20 min; or 2) isocratic (A–water with 0.05% formic acid and B–acetonitrile) of A:B = 95:5 consistently for 20 min.

### 2.2 1-Mode network analysis of herbs in prescriptions

We built a 1-mode network of herbs used in anti-obesity prescriptions. To collect information on herbal constituents in obesity prescriptions published online, we searched original research articles on the anti-obesity effects of herbal prescriptions. Nine herbal prescriptions were selected based on preference of clinical usage. To examine combinations of herbs used in anti-obesity prescriptions, herbal compositions of prescriptions (contains at least two herbs) were arranged in a binary matrix (2-mode network) with herbs in rows and prescriptions in columns, in which '0' indicated absence and '1' indicated presence [29]. These 2-mode networks were transformed into 1-mode herb x herb network using the methods suggested by Breiger [30]. Similarity between herbs was assessed by analyzing Jaccard coefficients with values ranging from 0 to 1 using XLSTAT Excel add-in software [31]. An herb network was built using herbs as nodes and undirected edges as relationships in Cytoscape version 3.8.2. Node sizes represented degrees and edge width in the herb network represents Jaccard coefficients between nodes, and node colors represented the frequency of herb appearances. A heatmap of unweighted Pearson's correlations was created using XLSTAT.

## 2.3. Acquisition of potential active ingredients and targets of EH-CS combinations using web databases

The traditional Chinese medicine systems pharmacology database and analysis platform (TCMSP, https://old.tcmsp-e.com/tcmsp.php (accessed on 29 June 2021)) was used as a repository to collect data about the ingredients and targets of EH-CS. Two ADME properties of drug likeness (DL) (≥0.18) and oral bioavailability (OB) (≥30%) were used to identify potential bioactive ingredients in each herb. Pubchem_CID was used to identify ingredients. Proteins targeted by each ingredient acquired from the TCMSP were validated using the Uniprot database (https://www.uniprot.org/ (accessed on 29 June 2021)) [32]. All target proteins were validated and converted into official gene names in Homo sapiens using the Genecards database ((https://www.genecards.org/) (accessed on 29 June 2021)) [33]. Lists of ingredients and targets were sorted and uploaded as groups for each herb to the bioinformatics and evolutionary genomics website (http://bioinformatics.psb.ugent.be/webtools/Venn/ (accessed on 20 July 2021)) to obtain Venn diagrams.

## 2.4. Key EH-CS target screening for obesity using the STRING database and Cytoscape

Targets related to obesity were obtained by selecting overlapping targets identified using the Genecards database (accessed on 10 July 2021) and the Disgenet web database (https://www.disgenet.org/ (accessed on 10 July 2021)) [34]. All targets of EH-CS ingredients were arranged as lists and uploaded to the STRING database (https://string-db.org/ (accessed on 13 July 2021)) [35] to construct a protein-protein interaction (PPI) network. The minimum required interaction score was set at 0.4, and isolated target nodes without known interactions were discarded. PPI network interactions were exported to Cytoscape [36], and the network file was imported into Cytoscape version 3.8.2; topological analysis of networks was performed using a built-in network analyzer. Two topological parameters, "degree" and "betweenness centrality" were adopted as criteria for selecting key targets of EH-CS combinations. Targets above average for degree and betweenness centrality were regarded as key targets.

## 2.5. Construction of a compound-target network

A network of herbal compound-targets was constructed and visualized in Cytoscape and processed in PowerPoint software. Nodes represent herb ingredients, and targets, and edges represent interactions between nodes. Node colors of compounds represent source herbs (CS-blue, EH-red, EH-CS-purple). Node colors of targets represent separate three clusters. The clusters of key targets were created using on-board function in STRING database according to the kmeans algorithm.

## 2.6. The KEGG pathway and gene ontology enrichment analyses in the R package

The Kyoto Encyclopedia of Genes and Genomes (KEGG) [37] pathway and gene ontology (GO) enrichment analyses [38] were performed by uploading key target genes to the DAVID database platform (https://david.ncifcrf.gov/ (accessed on 13 July 2021)) [39]. A list of all key genes was uploaded, and the identifier was set as "official gene symbol". Annotations of key genes were identified in the KEGG pathway and three GO terms, that is, BP (biological process), CC (cellular component), and MF (molecular function). The top 20 results with the lowest P-values in each category were visualized as a bubble chart containing P-values, gene

counts, and gene ratios. Bubble plots were created using R Studio and the 'ggplot2' package with public R script, which was modified for the present study [40].

## 2.7. Herbal extraction preparation

Five Ephedrae herba and Coicis semen (EH-CS) combinations were used in this study (0:100, 25:70, 50:50, 75:25, and 100:0 (w/w)). Herbs were obtained from Humanherb (Gyeongsang-bukdo, South Korea). Different ratios of EH to CS were extracted in hot water for 1 h at 95°C. Crude extracts were filtered, condensed using a rotary evaporator (Buchi, Switzerland) at 95°C, and freeze-dried to obtain respective powders, which were eluted with Dulbecco's Phosphate Buffered Saline (DPBS) and filtered through a 0.22 μm syringe filter before use.

## 2.8. Cell culture and cell differentiation

3T3-L1 preadipocytes (ATCC CL-173) were grown in Dulbecco's Modified Eagle Medium (DMEM, Gibco, Carlsbad, CA, USA), supplemented with 10% fetal bovine serum (FBS) and 100 U/mL penicillin and streptomycin (Gibco, USA). Cells were incubated at 37°C in a humidified 5% $CO_2$ atmosphere and maintained at ~70% confluence before being used in experiments.

3T3-L1 preadipocytes were seeded on 12-well plates at $2 \times 10^5$ cells per well in DMEM supplemented with 10% FBS and incubated to full confluence (100%) and then for a further 2 days. Differentiation was initiated by exchanging the medium with differentiation medium (DMEM supplemented with 10% FBS, 1 μM dexamethasone, 0.5 mM 3-isobutyl-1-methylxanthine, 10 μg/ml insulin) for 72 h, and cells were further incubated in maturation medium (DMEM supplemented with 10% FBS containing 10 μg/ml insulin) for 1 day (for real-time PCR) or 8 days (for ORO staining and Western blotting).

## 2.9 Cell viability assessment

For viability determinations we used previous protocol with slight modification [41]. Cells were seeded in 96-well plates in FBS-free DMEM at $2 \times 10^3$ cells per well and then incubated with various concentrations of EH-CS combinations for 24 h. Viabilities were measured using an Ez-Cytox kit (Daeil Lab., Seoul, South Korea) according to the manufacturer's instructions. Optical densities (ODs) were then measured at 450 nm using a microplate spectrophotometer (VersaMax, Molecular Devices, CA, USA).

## 2.10 Oil red O staining

For ORO staining, we used modified protocols from previous study [42]. Mature 3T3-L1 cells (adipocytes) were washed with DPBS, fixed with 5% formalin for 1 h at room temperature, washed once with 60% isopropanol, and dried. A stock solution of ORO was prepared by filtering a solution of 0.175 g of ORO powder in 50 ml of isopropanol and diluting the filtrate with distilled water at a ratio of 3: 2. Cells were stained with ORO solution for 15 min, washed with distilled water, air-dried, and examined under an inverted microscope system equipped with a camera (DMI 6000, Leica, Wetzlar, Germany). For quantitative analysis, stains were re-dissolved in isopropanol, and absorbances were measured at 520 nm using a spectrophotometer (VersaMax).

## 2.11 The palmitate-induced preadipocyte inflammatory model

3T3-L1 cells (preadipocytes) were seeded on 12-well plates without differentiation factors in culture medium and incubated for 24 h in PA containing medium for 24 h [43]. PA medium

was prepared by conjugating palmitate (0.5 mM) in 1% bovine serum albumin (BSA)-containing DMEM for 1 h at 55°C, as described by Kim et al. [44] with slight modification.

## 2.12 Real-time quantitative PCR

Total RNA was isolated from preadipocytes or adipocytes using Trizol reagent (Thermo Fisher Scientific, USA). Reverse transcription was performed using an AccuPower RT PreMix (Bioneer, Daejeon, South Korea) and oligo (dt) 18 primers (Invitrogen, Carlsbad, CA, USA). cDNA amplification was performed using a LightCycler 480 PCR system (Roche, Basel, Switzerland). PCR reaction mixes contained 10 μl of 2x SYBR Green Master Mix (Roche, Switzerland), 8 μl of ultrapure water, 10 pmol/μl of primers, and 1 μl of template cDNA. PCR was performed using an initial denaturation step (95°C for 10 min), 45 amplification cycles (denaturation at 95°C for 10 s, annealing at 56~62°C for 20 s, and extension at 72°C for 20 s). Melting curve analysis was performed at 95°C for 5 min for quality check. Threshold cycle (Ct) value was calculated to quantify PCR results. Relative expression levels were calculated by dividing gene Ct values by that of β-actin. All data were acquired using a LightCycler 480 instrument and software. Primers sequences are presented in Table 1.

## 2.13 Western blot

Preadipocyte protein levels were determined by Western blot. Briefly, cells were washed and lysed with radioimmunoprecipitation assay (RIPA) buffer (Thermo Fisher Scientific, Rockford, IL, USA) containing an enzyme inhibitor cocktail (Gendepot, Barker, TX, USA). Protein concentrations were estimated using the BCA kit (Thermo Fisher Scientific). Same amounts of protein lysates were loaded into 10% SDS-PAGE gels, electrophoresed, and transferred to

**Table 1. Primer sequence used in this study.**

| Genes | Sequence (5'-3') | | Tm (°C) |
|---|---|---|---|
| **PPARγ** | Forward | agtgacttggctatatttatagctgtcatt | 65.3 |
| | Reverse | tgtcttggatgtcctcgatgg | 61.3 |
| **FABP4** | Forward | cagaagtgggatggaaagtcg | 61.3 |
| | Reverse | cgactgactattgtagtgtttga | 59.3 |
| **CEBPA** | Forward | gcgcaagagccgagataaag | 60.5 |
| | Reverse | cacggctcagctgttcca | 58.4 |
| **SCD1** | Forward | atatcctggtttccctgggt | 58.4 |
| | Reverse | cagcggtactcactggc | 57.2 |
| **FASN** | Forward | cctccaagactgactcgg | 58.4 |
| | Reverse | cagtgtgctcaggttcagtt | 58.4 |
| **ACC1** | Forward | tggcgtccgctctgtgata | 59.5 |
| | Reverse | catggcgacttctgggttg | 59.5 |
| **SREBF1** | Forward | ggaacagacactggccga | 58.4 |
| | Reverse | aagtcactgtcttggttgttgat | 59.3 |
| **PTGS2** | Forward | gcgacatactcaagcaggagca | 64 |
| | Reverse | agtggtaaccgctcaggtgttg | 64 |
| **IL6** | Forward | ccacttcacaagtcggaggctta | 64.7 |
| | Reverse | gcaagtgcatcatcgttgttcatac | 64.2 |
| **TNF** | Forward | aagcctgtagcccacgtcgta | 63.3 |
| | Reverse | ggcaccactagttggttgtctttg | 65.3 |
| **β-Actin** | Forward | gacggccaggtcatcactattg | 64 |
| | Reverse | ccacaggattccatacccaaga | 62.1 |

PVDF membranes using an electrophoretic transfer cell (Bio-rad, Hercules, CA, USA). Membranes were then blocked with 5% BSA in TBS-T (TBS containing 0.1% Tween 20) for 2 h at room temperature, incubated with primary antibodies (1:1000 dilution in TBS-T) overnight with gentle agitation, rinsed with TBS-T, and incubated with secondary antibodies (1:3000 dilution in TBS-T) at room temperature for 2 h. Chemiluminescent blots were developed using ECL buffer (Super Signal West Pico, Thermo Fisher Scientific), and images were captured using the Fusion Solo imaging system (Vilber Lourmat, France).

## 2. 14 Elisa

Concentrations of secretory inflammatory cytokines in cell culture supernatants were measured using Quantikine mouse ELISA kits (R&D Systems, Inc. Minneapolis, MN, USA). Briefly, conditioned media of palmitate-induced preadipocytes (Section 4.10) and IL-6 concentrations were determined according to the manufacturer's instructions. Optical densities were measured at 450 nm using a microplate spectrophotometer (VersaMax).

## 2.15 Statistical analysis

The significances of differences between non-treated 3T3-L1 cells and differentiated cells and between sample-treated differentiated cells and differentiated cells were determined by one-way ANOVA in Graphpad Prism 5.0 (Graphpad Software, USA). Results are presented as the means ± SDs of at least three independent experiments, and statistical significance was accepted for $P$-values $< 0.05$. Figures and tables were created using Graphpad Prism 5.0.

## 3. Results

### 3.1 Profiling and identification of major compounds in EH-CS by HPLC

All EH-CS samples were analyzed by HPLC fingerprinting using standard compounds. The peaks of HPLC chromatogram identified as a retention time of 1.070 min and 17.373 min which corresponded to ephedrine (Fig 1A) and stigmasterol (Fig 1B). Ephedrine was detected only in EH-containing samples (s2, s3, s4, s5). Meanwhile, Stigmasterol content was detected in s3, s4, s5.

### 3.2 Analysis of the Herb-Herb 1 mode network

The 1-mode Herb network revealed a higher frequency of Ephedrae herba (EH), Coicis semen (CS), and Glycyrrhizae Radix (GR) use in anti-obesity prescriptions (Fig 2A). High degree and betweenness centrality figures of EH and CS in herb network demonstrates their importance in these prescriptions. Unexpectedly, Jaccard similarity coefficiency results between herbs based on their frequencies showed a higher value for minor herbs due to their consistent absence in most prescriptions. Pearson's correlation matrix showed hierarchical distance between two clusters (CS and EH cluster), which is relatively far apart (Fig 2B). We speculate that this result was caused by reference article sampling method only limited to anti-obesity herbal prescriptions, which has low coexistence in between EH and CS.

### 3.3 Active compound screening and key targets of EH-CS in combination

TCMSP database showed that 15 and 7 ingredients in EH and CS, respectively, met OB and DL criteria (Table 2). Mandenol and stigmasterol were common constituents of EH and CS. (Fig 3A, Table 2).

Lists of potential targets of active ingredients from EH and CS were obtained and compared with lists of obesity-related targets obtained from web databases (Fig 3B). As a result, 2199 targets were identified from disease databases, as compared with 223 EH targets and 31 CS

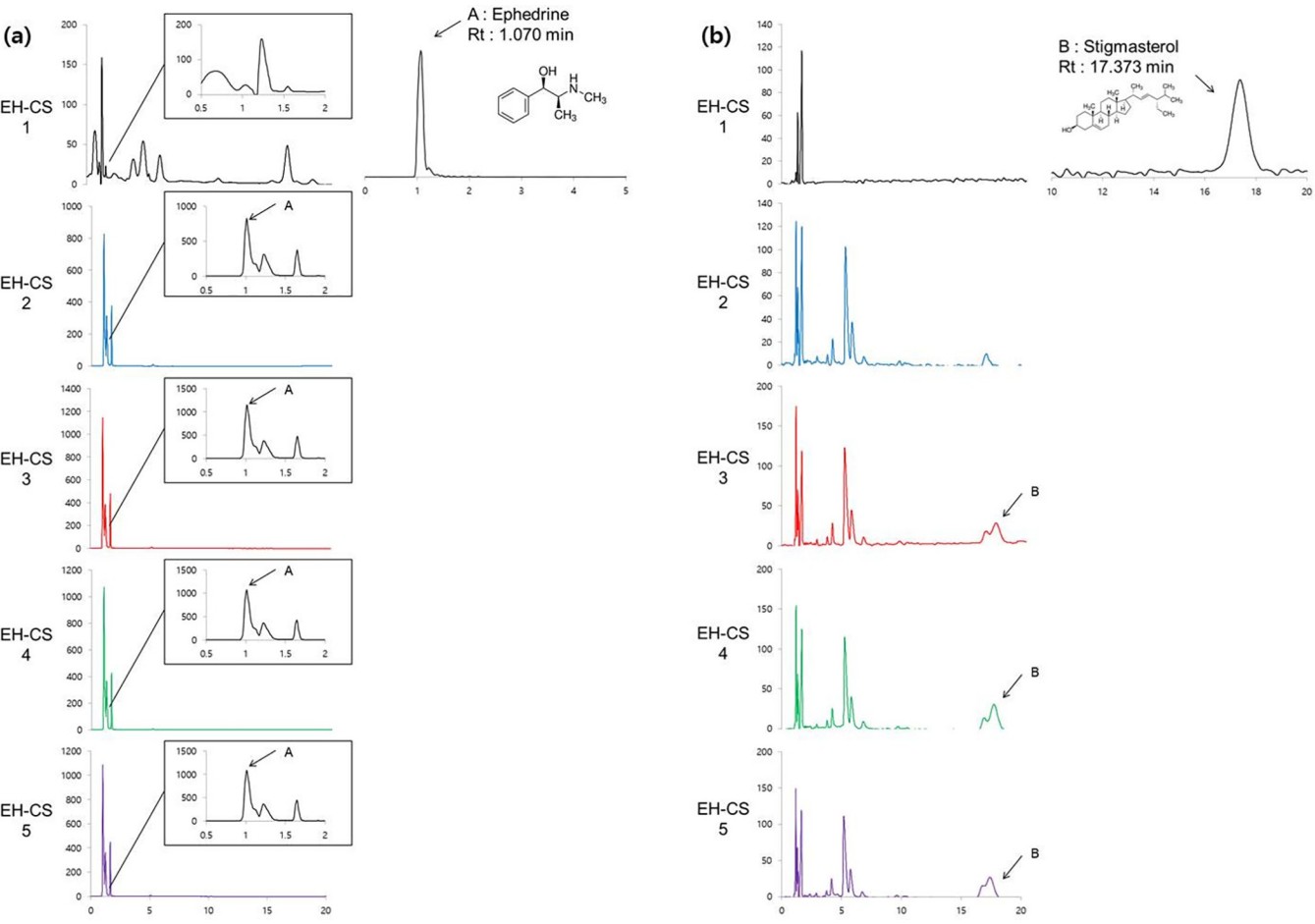

**Fig 1.** (a) HPLC analysis results for EH-CS samples and standard (Ephedrine) in isocratic condition (A–water with 0.05% formic acid and B–acetonitrile) of A:B = 50:50. Ephedrine was identified at a retention time of 1.070 min. (b) HPLC analysis results for EH-CS samples and standard (Stigmasterol) in isocratic condition (A–water with 0.05% formic acid and B–acetonitrile) of A:B = 95:5. Stigmasterol was identified at a retention time of 1.737 min.

targets. Interestingly, the 31 CS targets were all included among EH targets. Finally, total 152 genes targeted by EH and CS were found to be related to obesity.

## 3.4 Construction of a PPI (protein-protein interaction) network and screening of key targets

The 152 obesity-related target genes were uploaded into the STRING database to obtain a PPI network, which was then analyzed in Cytoscape. Network parameters of betweenness centrality and degree were evaluated. Seventeen solitary proteins without a known interaction were removed. Finally, 31 target proteins were found to be highly interconnected and thus were selected as key EH-CS targets. A list of key targets arranged by degree and betweenness centrality is provided in Table 3. Fig 4 shows the PPI network consisted of 31 key targets as nodes and 308 interactions between targets as edges. IL6 was found to be a core target gene of the PPI network with the highest degree of 28 (Table 3).

## 3.5. KEGG pathway and GO enrichment analysis of the key target genes

We uploaded 31 key genes into the DAVID database and obtained 20 results for the KEGG pathway and GO enrichment analysis using smallest *P*-values (Fig 5). Of the 20 pathways, 11

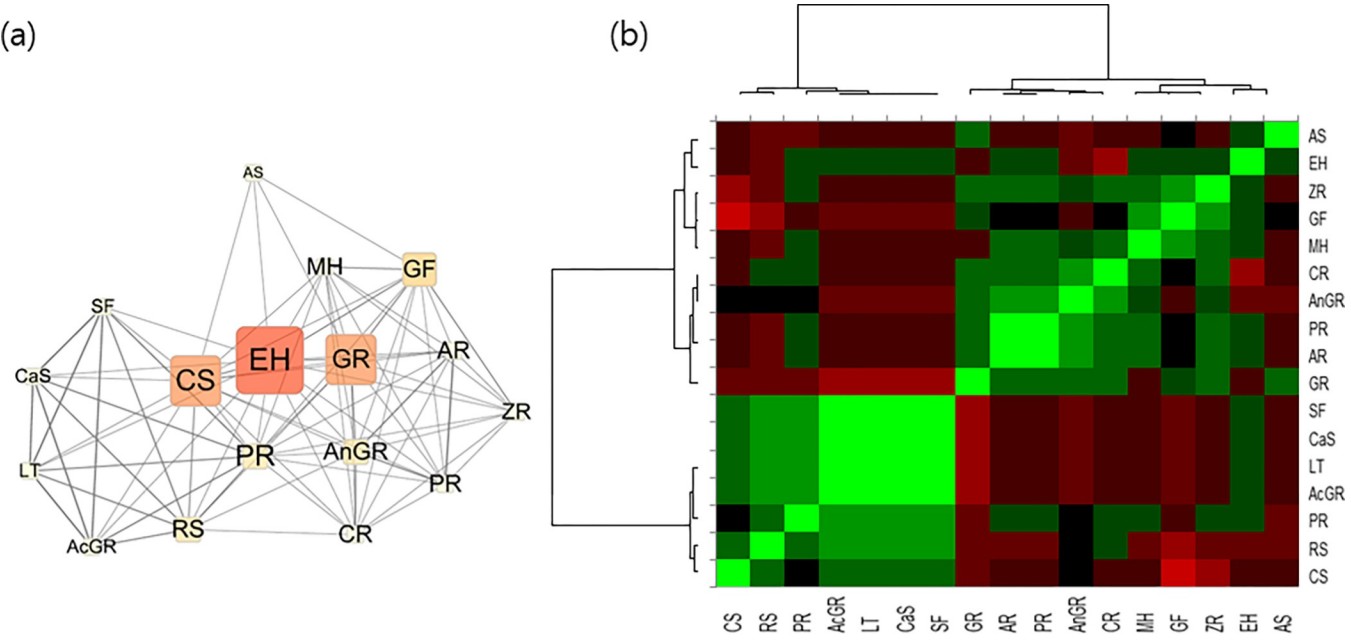

**Fig 2.** (a) 1-mode network of herbs in herbal prescriptions (b) Heatmap of Pearson's correlation matrix of herbs in anti-obesity herbal prescriptions. AS, Armeniacae Semen; EH, Ephedrae Herba; ZR, Zingiberis Rhizoma; GF, Gypsum Fibrosum; MH, Menthae Herba; CR, Cnidii Rhizoma; AnGR, Angelicae Gigantis Radix; PR, Paeoniae Radix; AR, Atractylodis Rhizoma; GR, Glycyrrhizae Radix; SF, Schisandrae Fructus; CaS, Castaneae Semen; LT, Liriope Tuber; AcGR, Acori Graminei Rhizoma; PR, Platycodonis Radix; RS, Raphani Semen; CS, Coicis Semen.

were human disease pathways and 9 signaling pathways. Of the 9 signaling pathways, 5 pathways were involved in signal transduction, 2 in the immune system, and 2 in the endocrine system. Among the signaling pathways, TNF and thyroid hormone signaling pathways were most significantly enriched.

For BP terms, drug response and positive regulation of transcription from RNA polymerase II promoter were the most prominent results. For CC terms, nucleoplasm was the most significant term, followed by protein-containing complex, and for MF terms, enzyme binding and protein-containing complex binding were the most significant. Target genes related to the most prominent BP, CC, and MF terms were annotated and colored red, blue, and green, respectively, in the PPI network (Fig 4).

### 3.6 Compound-target network and target clustering of EH-CS

Using the key targets of EH-CS elicited by PPI network analysis, we constructed 2-dimensional compound-target network (Fig 6). The complete compound-target network consisted of 45 nodes and 81 edges. EH and CS contained 13 and 3 active compounds, respectively, and 2 compounds were shared (stigmasterol and mandenol). Thirty-one targets were linked with EH compounds, 3 of which were also linked with CS compounds. However, no key target was solely linked with CS. The three common key targets shared by the two herbs were Prostaglandin Endoperoxide Synthase 2 (PTGS2, also called COX2), Nuclear Receptor Coactivator 1 (NCOA1), and Adrenoreceptor Beta 2 (ADRB2).

Key targets were clustered into 3 groups (cluster 1–3), and their significant BP terms are presented (Table 4). In cluster 1, cellular response to reactive oxygen species was most prominent BP term. In cluster 2, positive regulation of apoptotic process was second most significant BP term following the response to glucocorticoid. In cluster 3, positive regulation of protein

**Table 2. List of potential active compounds in Ephedrae herba and Coicis semenSource.**

| | Molecular name | Pubchem CID | MW | OB | DL |
|---|---|---|---|---|---|
| CS/EH Compound (2) | Mandenol | 5282184 | 308.56 | 42.0 | 0.19 |
| | Stigmasterol | 5280794 | 412.77 | 43.83 | 0.76 |
| EH compound (15) | Eriodictyol | 440735 | 288.27 | 71.79 | 0.24 |
| | Truflex OBP | 66540 | 334.5 | 43.74 | 0.24 |
| | Genkwanin | 5281617 | 284.28 | 37.13 | 0.24 |
| | Naringenin | 932 | 272.27 | 59.29 | 0.21 |
| | Beta-sitosterol | 222284 | 414.79 | 36.91 | 0.75 |
| | Quercetin | 5280343 | 302.25 | 46.43 | 0.28 |
| | Herbacetin | 5280544 | 302.25 | 36.07 | 0.27 |
| | Clionasterol (gamma-sitosterol) | 457801 | 414.79 | 36.91 | 0.75 |
| | Campesterol | 173183 | 400.76 | 37.58 | 0.71 |
| | Kaempferol | 5280863 | 286.25 | 41.88 | 0.24 |
| | 24-Ethylcholest-4-en-3-one | 15596633 | 412.77 | 36.08 | 0.76 |
| | Pectolinarigenin | 5320438 | 314.31 | 41.17 | 0.3 |
| | Supraene | 638072 | 410.8 | 33.55 | 0.42 |
| | Luteolin | 5280445 | 286.25 | 36.16 | 0.25 |
| | Diosmetin | 5281612 | 300.28 | 31.14 | 0.27 |
| CS compound (7) | Coixenolide | 46173943 | 591.08 | 32.4 | 0.43 |
| | Hydrosqualene | 11975273 | 410.8 | 33.55 | 0.42 |
| | 2-Monoolein | 5319879 | 356.61 | 34.23 | 0.29 |
| | Sitosterol alpha1 | 9548595 | 426.8 | 43.28 | 0.78 |
| | CLR (Cholesterol) | 5997 | 386.73 | 37.87 | 0.68 |
| | Monooleoylglycerol | 11451146 | 356.61 | 34.13 | 0.3 |
| | Sitosterol (3-epi-beta-sitosterol) | 12303645 | 414.79 | 36.91 | 0.75 |

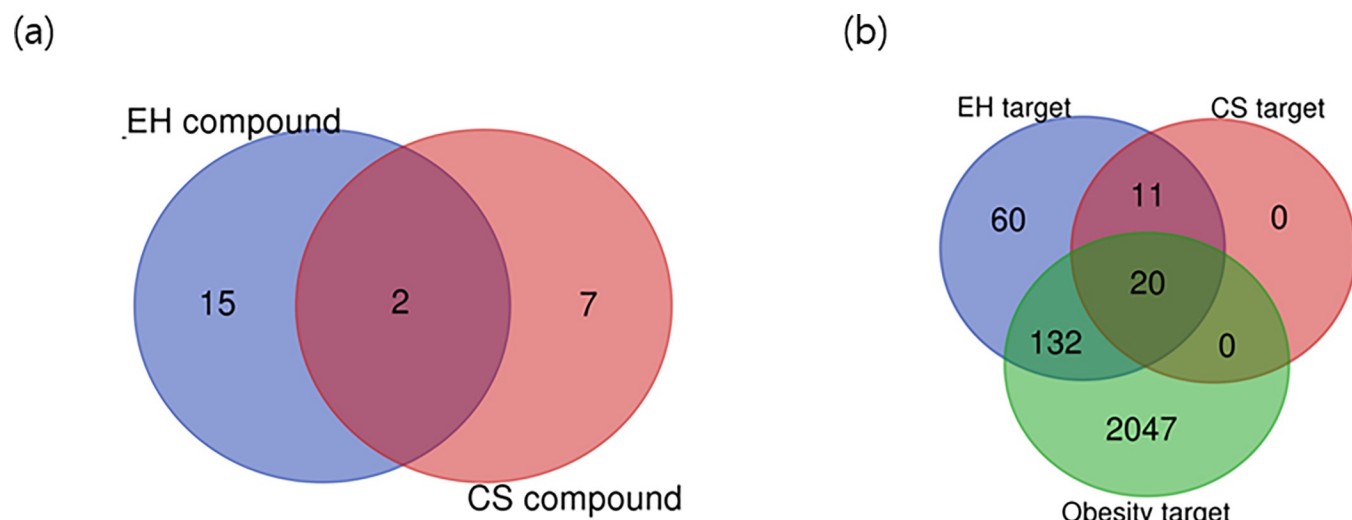

**Fig 3.** (a) Venn diagram showing the distribution of potent active compounds in EH-CS combinations (b) Venn diagram showing all the targets of EH and CS related to obesity. Data were obtained from TCMSP.

kinase B signaling, positive regulation of phosphorylation, and positive regulation of MAP kinase activity were the three most important BP terms.

## 3.7 Cytotoxicity of EH-CS samples on preadipocytes

3T3-L1 cells were treated with EH-CS of five different EH to CS ratios (s1-s5) at concentrations from 0 to 50 μg/ml (Fig 7A). Sample s1 (100% CS extract) at 50 μg/ml did not have any significant cytotoxic effect on preadipocytes. However, as the EH ratio increased, significant cytotoxicity was observed; for example, s5 (100% EH) reduced cell viability to ~90% at 30 μg/ml. Thus, we decided to use a maximum concentration of 25 μg/ml in subsequent studies.

## 3.8 Inhibitory effects of EH-CS on adipocyte differentiation

EH-CS treatments of preadipocytes at 25 μg/ml for 8 days significantly inhibited differentiation, as demonstrated by loss of lipid accumulation in Oil Red O (ORO) staining (Fig 7B). In particular, s3 (EH:CS = 1:1 (w/w)) reduced lipid accumulation most. Pravastatin (50 μg/ml) was used as a positive control.

**Table 3. Degrees and betweenness centralities of the 31 key target genes targeted by EH-CS as predicted by the PPI network structure.**

| Gene name | Target name | Degree | Betweenness Centrality |
|---|---|---|---|
| IL6 | Interleukin 6 | 28 | 0.034933 |
| TNF | Tumor Necrosis Factor | 27 | 0.029773 |
| AKT1 | AKT Serine/Threonine Kinase 1 | 26 | 0.02412 |
| ESR1 | Estrogen Receptor 1 | 26 | 0.044226 |
| TP53 | Tumor Protein P53 | 26 | 0.022326 |
| VEGFA | Vascular Endothelial Growth Factor A | 26 | 0.017908 |
| EGFR | Epidermal Growth Factor Receptor | 25 | 0.015777 |
| MAPK3 | Mitogen-Activated Protein Kinase 3 | 25 | 0.014045 |
| PTGS2 | Prostaglandin-Endoperoxide Synthase 2 | 25 | 0.03103 |
| EGF | Epidermal Growth Factor | 24 | 0.006448 |
| HSP90AA1 | Heat Shock Protein 90 Alpha Family Class A Member 1 | 24 | 0.011566 |
| PPARA | Peroxisome Proliferator Activated Receptor Alpha | 24 | 0.02819 |
| CASP3 | Caspase 3 | 23 | 0.00302 |
| FOS | Fos Proto-Oncogene, AP-1 Transcription Factor Subunit | 23 | 0.010306 |
| MYC | MYC Proto-Oncogene, BHLH Transcription Factor | 23 | 0.00302 |
| CCND1 | Cyclin D1 | 22 | 0.009491 |
| MAPK1 | Mitogen-Activated Protein Kinase 1 | 22 | 0.004374 |
| MAPK8 | Mitogen-Activated Protein Kinase 8 | 22 | 0.0021 |
| STAT1 | Signal Transducer And Activator Of Transcription 1 | 21 | 0.001201 |
| APP | Amyloid Beta Precursor Protein | 20 | 0.007925 |
| CASP8 | Caspase 8 | 20 | 6.54E-04 |
| IL4 | Interleukin 4 | 19 | 0.00576 |
| PRKCA | Protein Kinase C Alpha | 17 | 1.04E-04 |
| IGFBP3 | Insulin Like Growth Factor Binding Protein 3 | 16 | 0.002014 |
| CYP3A4 | Cytochrome P450 Family 3 Subfamily A Member 4 | 13 | 0.019348 |
| F2 | Coagulation Factor II, Thrombin | 13 | 0.007026 |
| CYP1A1 | Cytochrome P450 Family 1 Subfamily A Member 1 | 10 | 0.006373 |
| APOB | Apolipoprotein B | 8 | 5.92E-04 |
| ADRB2 | Adrenoceptor Beta 2 | 7 | 4.47E-04 |
| NCOA1 | Nuclear Receptor Coactivator 1 | 7 | 0.00119 |
| AKR1C3 | Aldo-Keto Reductase Family 1 Member C3 | 4 | 2.30E-04 |

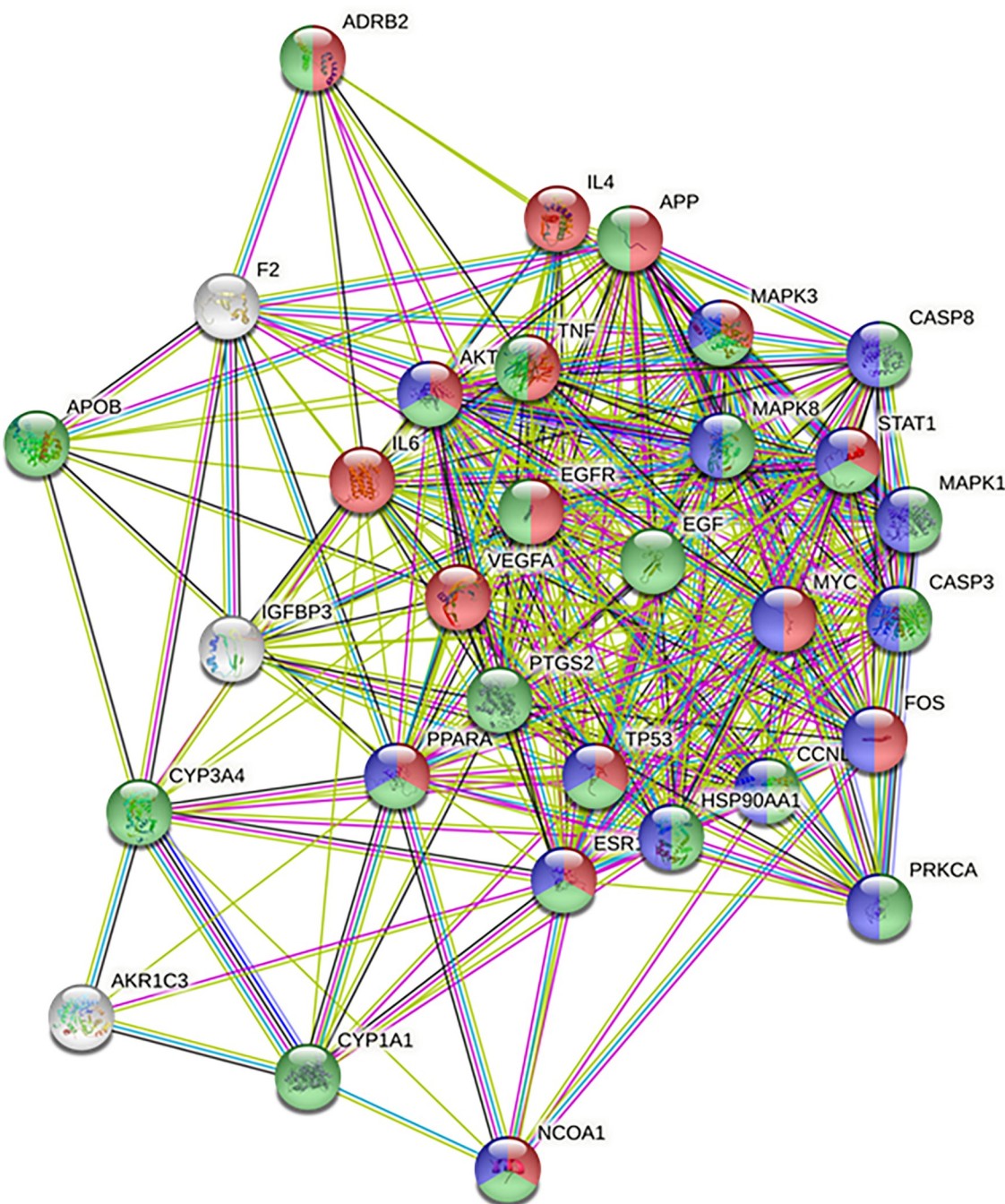

**Fig 4. PPI network of the 31 key targets of EH-CS.** Thirty-one nodes and 308 interactions between nodes are represented. Red nodes represent genes with dominant BP terms (positive regulation of transcription from RNA), blue nodes represent genes with dominant CC terms (nucleoplasm), and green nodes represent genes with dominant MF terms (enzyme binding).

## 3.9 Regulatory effects of EH-CS on adipocyte differentiation and lipogenesis markers

Treatment-induced changes in phosphorylated AMPK levels (an important modulator of differentiation and energy expenditure) in adipocytes were investigated by western blot. As

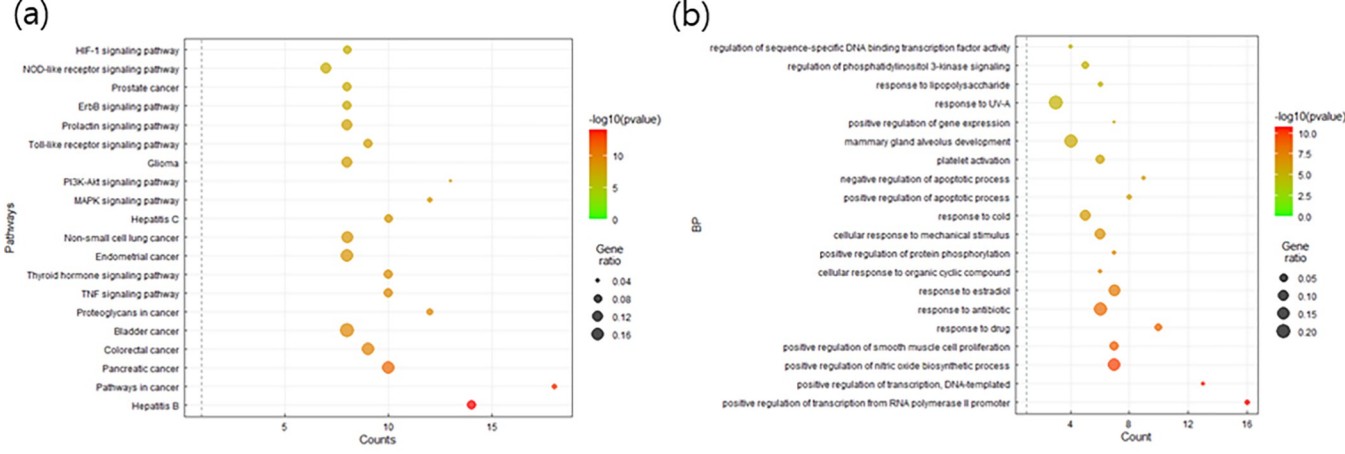

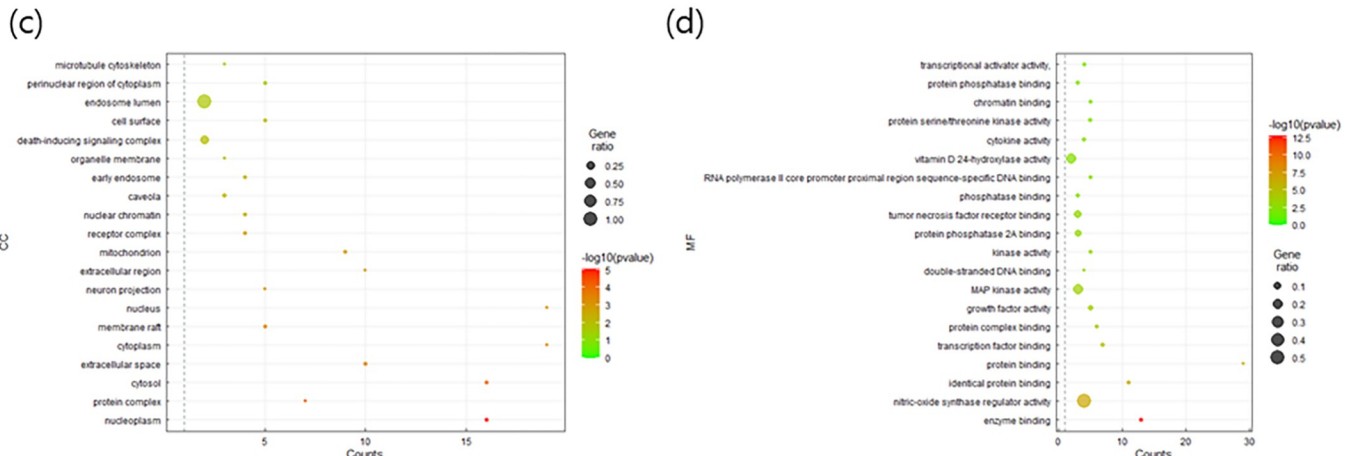

**Fig 5.** (a) Gene enrichment analysis of the KEGG pathway Gene ontology (GO) enrichment analysis of (b) biological processes, (c) cellular components, and (d) molecular function of key targets of EH-CS.

illustrated in Fig 8A and 8B, AMPK phosphorylation/AMPK was significantly increased by s3 and s5 as compared with that of differentiated adipocytes.

Real-time PCR was conducted to confirm the inhibitory properties and investigate the mechanisms responsible for the adipogenic and lipogenic effects of EH-CS in preadipocytes (Figs 8 and 9). At the early stage of differentiation (24 h), adipocytes showed significant increases in the expressions of the PPAR, FABP4, and CEBP genes, which ranged from 10 to 700-fold. However, treatment with EH-CS samples markedly reduced these increases. In particular, sample s3 potently inhibited the expressions of these adipogenic genes.

The lipogenic genes investigated were SCD1, FASN, ACC1, and SREBF1 (Fig 9). The expressions of these four genes were significantly increased during differentiation, but these increases were suppressed by EH-CS treatment, but not in a consistent manner. SCD1 and FASN (markers of lipogenesis) were significantly down-regulated by all five samples. However, the gene expressions of ACC1 and SREBF responded to EH-CS treatments inconsistently (Fig 9C and 9D), and s2 and s3 caused no significant change in therapeutic expressions.

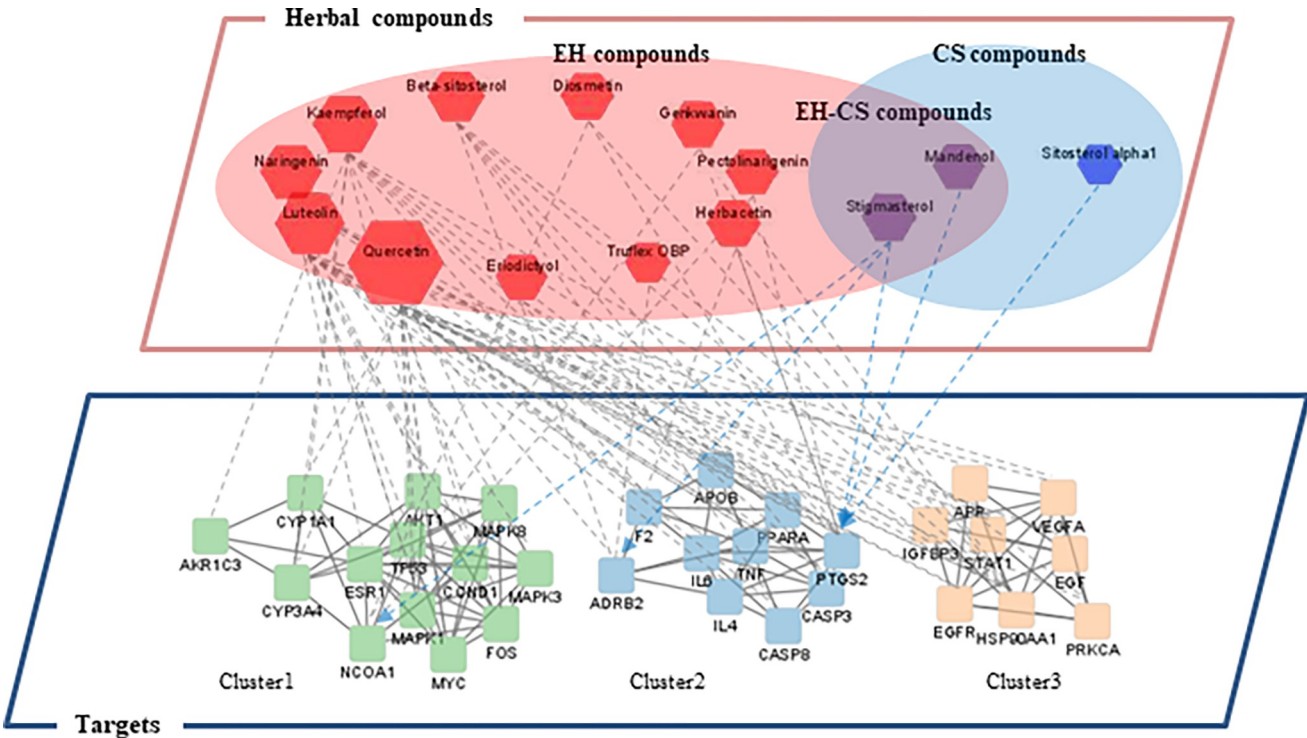

**Fig 6. Visualization of the compound-target network of EH-CS.** The upper layer represents herbal compounds from herbs and the lower layer represents networks of three modules (clusters) of 31 potential targets. Target clusters were created using kmeans clustering tools in STRING database.

### 3.10 Relative protein levels and gene expression levels of core targets of EH-CS combinations

The synergistic effects of EH-CS combinations on obesity were not fully explained by the expressional changes of lipogenic genes. As we predicted it as key modulator in network pharmacology analysis, we examined IL6, TNF, and PTGS2 (COX-2) gene changes in palmitate-stimulated (0.5 mM) preadipocytes incubated with the five EH-CS samples at 10 or 25 μg/ml (Fig 10). The expressions of these key markers were significantly upregulated by 5-, 3, and 7-fold, respectively by palmitate. Palmitate-induced increases in IL6 and PTGS2 gene expressions were significantly inhibited by most samples, except IL6 by s4 and PTGS2 by s5 at 25 μg/ml (Fig 10A and 10D), while s3 at 25 μg/ml had the strongest inhibitory effect. Interestingly, in

**Table 4. Description of Top 3 biological processes of target clusters within 31 key targets of EH-CS compounds for obesity as determined using STRING database.**

| Cluster | GO identifier | BP Term | Gene count | *P*-value |
|---------|--------------|---------|-----------|-----------|
| Cluster 1 | GO:0034614 | cellular response to reactive oxygen species | 6 | 1.60E-11 |
| | GO:0071276 | cellular response to cadmium ion | 6 | 2.70E-11 |
| | GO:0006974 | cellular response to DNA damage stimulus | 6 | 4.30E-07 |
| Cluster 2 | GO:0051384 | response to glucocorticoid | 4 | 3.50E-06 |
| | GO:0043065 | positive regulation of apoptotic process | 5 | 9.70E-06 |
| | GO:0032355 | response to estradiol | 4 | 1.60E-05 |
| Cluster 3 | GO:0051897 | positive regulation of protein kinase B signaling | 5 | 6.40E-08 |
| | GO:0042327 | positive regulation of phosphorylation | 4 | 1.30E-07 |
| | GO:0043406 | positive regulation of MAP kinase activity | 4 | 2.40E-06 |

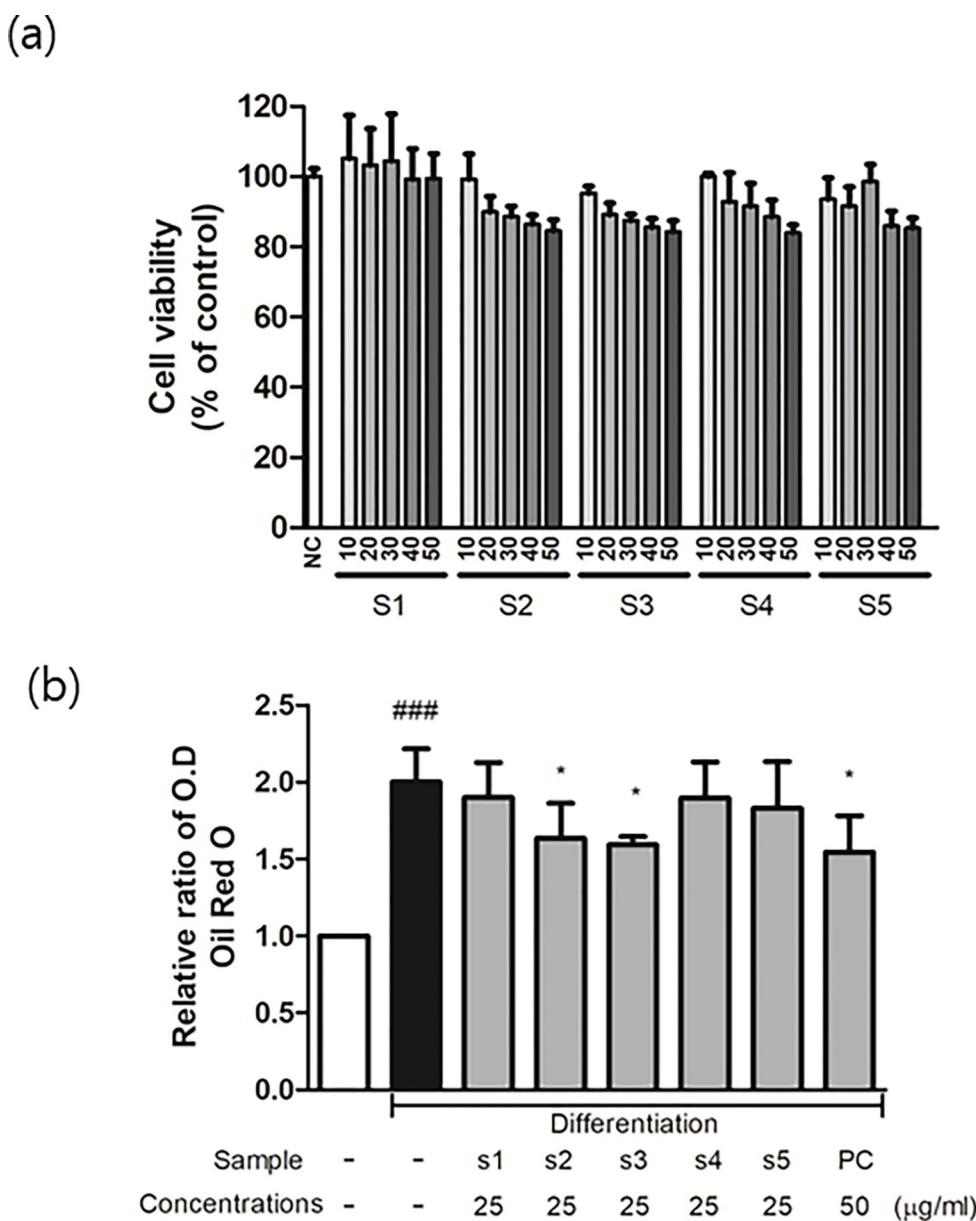

**Fig 7. Effect of EH-CS combinations on 3T3-L1 preadipocytes and lipid contents of mature adipocytes.** (a) Cell viabilities after treatment with EH-CS combinations (b) Oil Red O (ORO) staining results. Results are presented as the means ± SD of five different experiments. ###$P < 0.001$ versus non-differentiated 3T3-L1 preadipocytes, and *$P < 0.05$ versus differentiated adipocytes.

contrast to other samples (s1-s3), EH enriched samples (s4 and s5) at 25 μg/ml increased pro-inflammatory mediators within the cell viability range.

## 4. Discussion

Synergy is a commonly used concept in herbal medicine which can have advantages over single compound-based treatments [45]. By definition, synergy is said to exist when the combined effect of constituents is greater than the effect expected by summing their individual effects [46]. The rationale of combination therapy is that a drug combination has a greater effect than

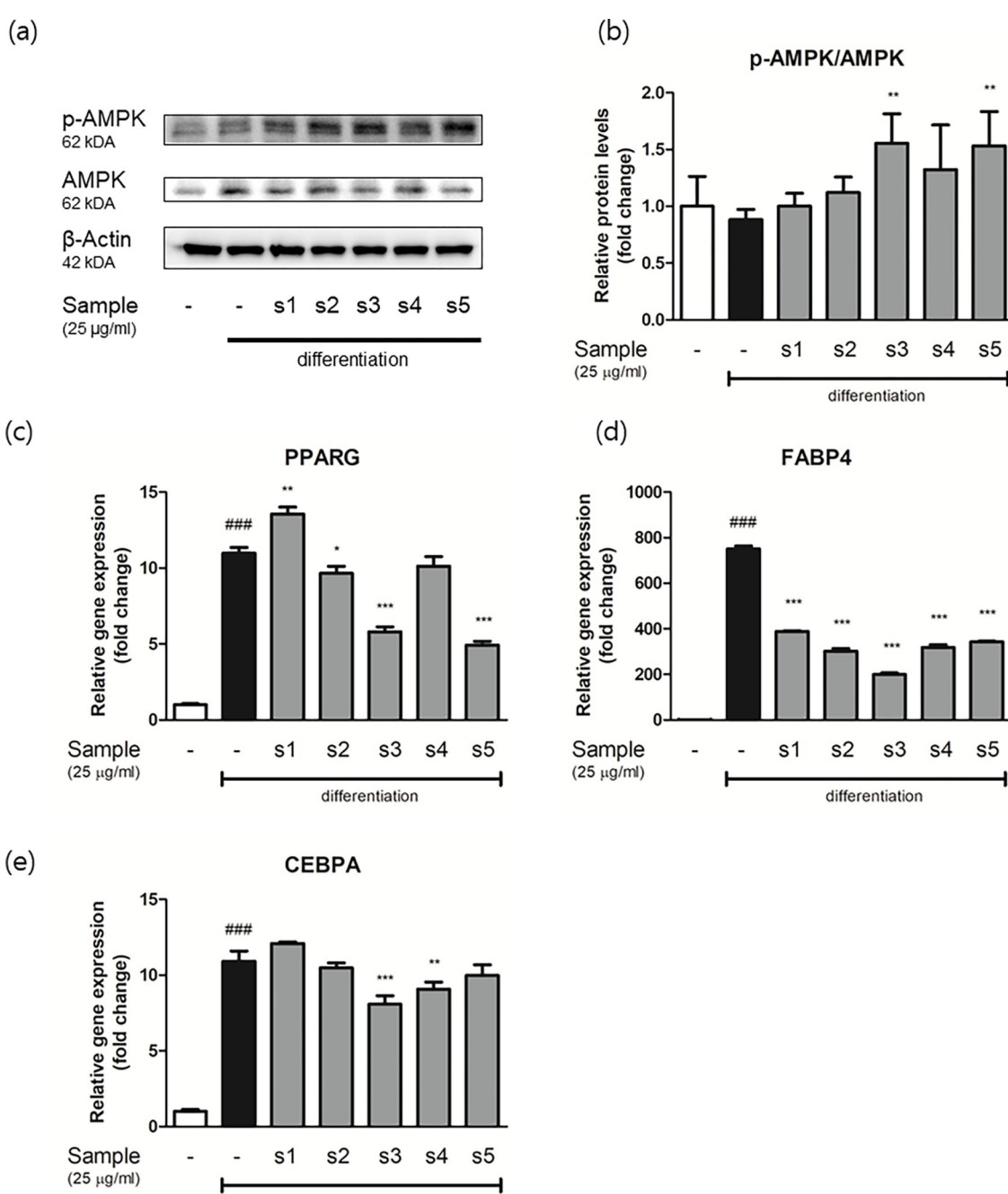

**Fig 8. Relative protein and gene expression levels of adipocytes differentiation markers as determined by western blot and real-time quantitative PCR in adipocytes.** (a) Representative blot of p-AMPK and AMPK. (b) Band intensities were measured densitometrically and divided by that of non-phosphorylated AMPK. Real-time quantitative PCR results for (c) PPAR gamma, (d) FABP4, and (e) C/EBPa. Relative gene expressions were calculated by dividing each Ct value by that of β-actin. Results are presented as the means ± SDs of three different experiments. ###$P < 0.001$ versus preadipocytes, and *$P < 0.05$, **$P < 0.01$, and ***$P < 0.001$ versus differentiated adipocytes.

single drug components by targeting multiple nodes in pathological pathways to overcome disease [47].

As herbal extracts are complex mixtures of numerous compounds, it is extremely difficult to predict their pharmaceutical potentials [48]. Furthermore, the synergistic and antagonistic

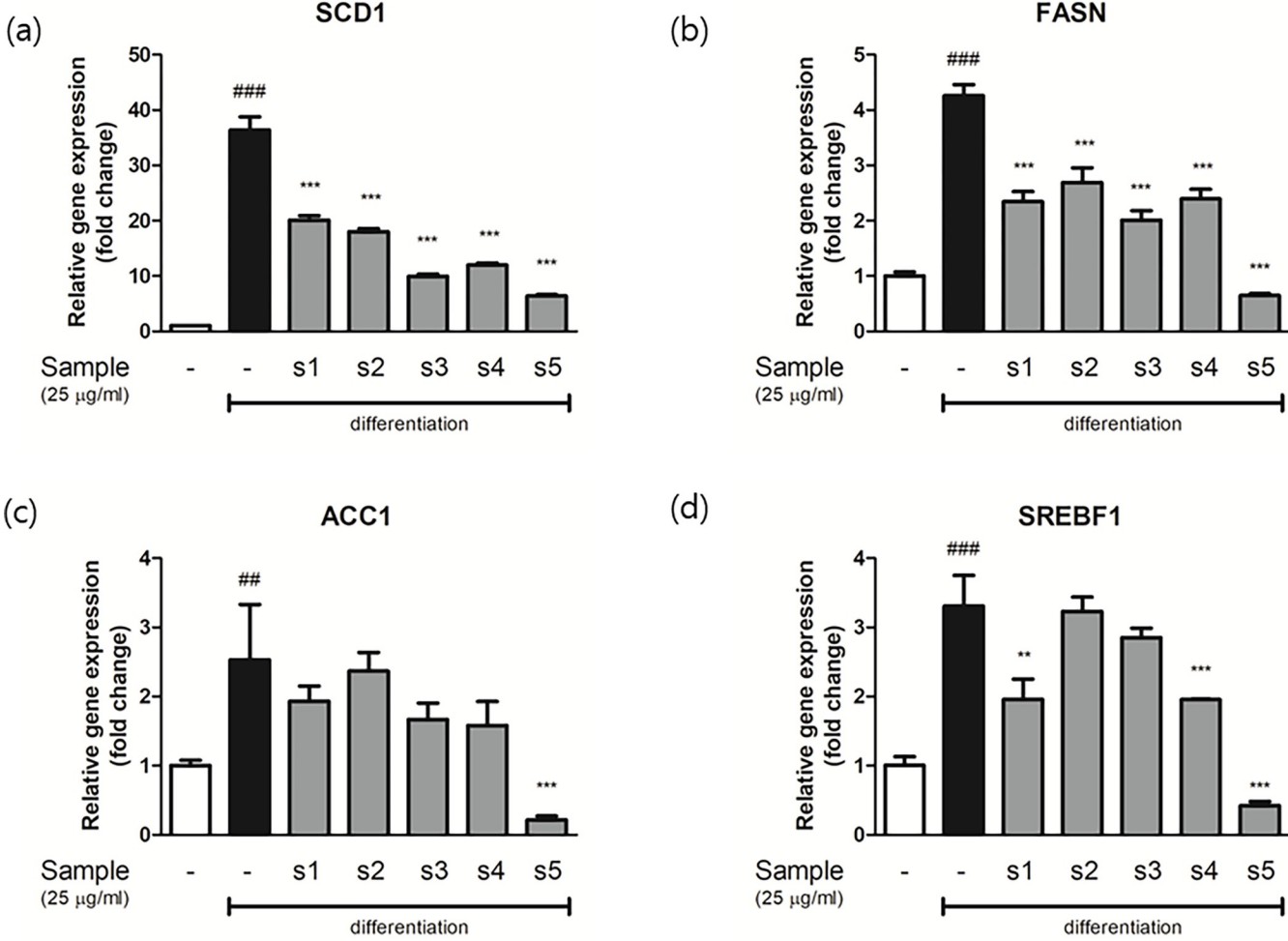

**Fig 9. Relative gene expression levels of lipid synthesis markers were analyzed by real-time quantitative PCR in adipocytes.** Real-time quantitative PCR results of (a) SCD1, (b) FASN, (c) ACC1, and (d) SREBF1. Relative gene expressions were calculated by dividing Ct values by that of β-actin. Results are presented as the means ± SD of three different experiments. ##$P < 0.01$ and ###$P < 0.001$ versus 3T3-L1 preadipocytes, **$P < 0.01$, and ***$P < 0.001$ versus differentiated adipocytes.

effects of herbal extracts are even more difficult to study because these effects involve considerations of interactions between numerous constituents [46]. Network pharmacology based on omics tools and web pharmaceutical databases provides a novel means of systematically analyzing complex interactions between compounds and their biological functions. For instance, Zhang et al. predicted synergistic effects between ingredients of a herbal combination in a TCM formula for rheumatoid arthritis using a self-developed network pharmacology platform [49], and subsequently suggested that drug synergism might be the result of modulation of a feedback loop in the network [50].

Another group screened for effective combinations of herbal medicines and scrutinized their modes of action for the treatment of endometriosis using network pharmacology and data mining approaches [51]. This technique involves combining numerous databases and computational tools and provides a feasible means of understanding the effects of herbal combinations [52], empowering molecular fundamental for modernization of herbal prescription.

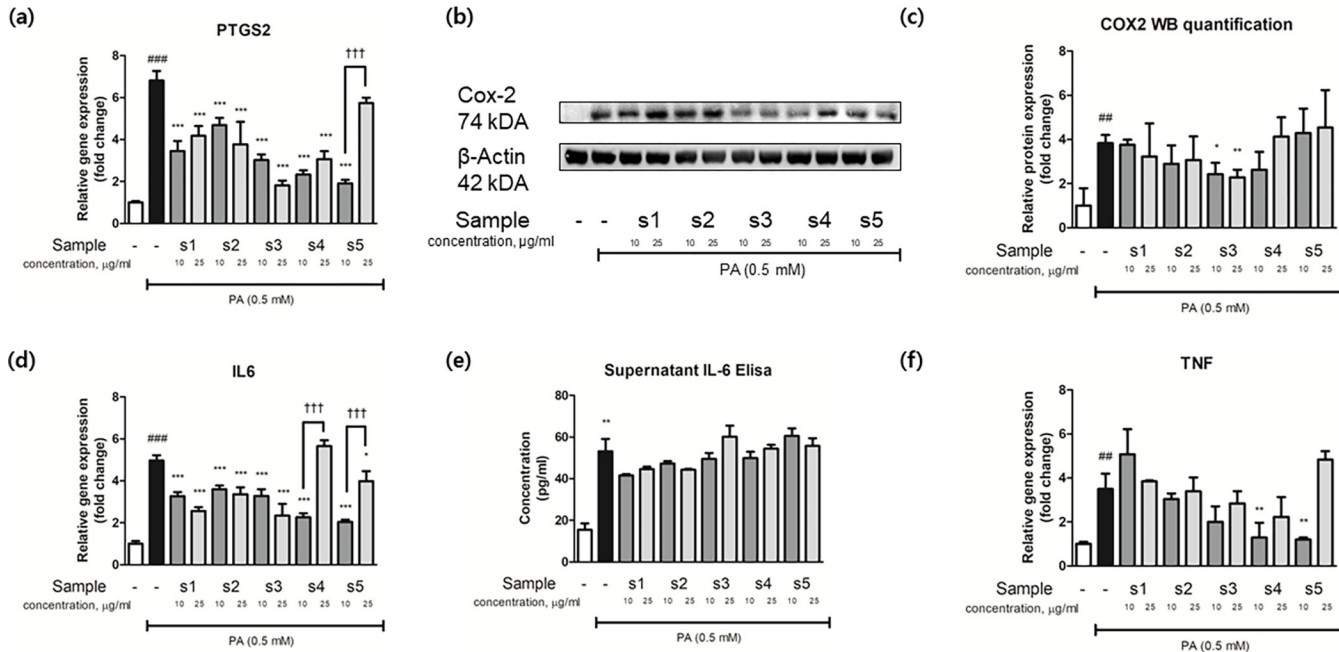

**Fig 10. Relative protein levels and gene expression levels of inflammatory markers, predicted by network pharmacology analysis, were subjected to western blot and real-time quantitative PCR in palmitate-treated (0.5 mM) preadipocytes.** Real-time quantitative PCR results for (a) PTGS2, (d) IL6, and (f) TNF. (b) Representative blots of PTGS2 and β-actin. (c) Band intensities were measured densitometrically and divided by that of β-actin. Relative gene expressions were calculated by dividing Ct values by that of β-actin. Results are presented as the means ± SDs of three different experiments. ##P < 0.01, and ###P < 0.001 versus non-treated 3T3-L1 preadipocytes, and *P < 0.05, **P < 0.01, and ***P < 0.001 versus non-treated adipocytes.

In the current study, the anti-obesity effects and mechanisms of different EH-CS samples on adipocytes were scrutinized. The study shows that: 1) EH extract has a greater impact on adipocyte viabilites thus limiting maximal concentration *in vitro*, 2) CS extract alone is not effective at reducing lipid accumulation in adipocytes in the long-term but does not reduce cell viability, and 3) EH-CS samples, especially s3 (EH-CS = 50:50), appear to have advantages over samples with higher EH ratios (s4, s5) in terms of efficacy and cell stability.

At a concentration of 25 μg/ml EH-CS combinations had variable effects on the expressions adipogenic genes, such as PPAR gamma, FABP4, and CEBPα, that were similar to or greater than those of EH (s5) (Fig 8). Their effects on phosphorylated AMPK levels suggested that s3 and s5 had similar inhibitory impacts on adipocyte differentiation. However, neural lipid contents in mature adipocytes were only significantly reduced by s3 (Fig 7B). There was a study that reports insulin-mimetic effects of CS, therefore, the adiposity of mature 3T3-L1 was increased by CS treatment on the contrary [53]. Regarding the EH, there was no reliable study that reports the efficacy of sole EH extract on 3T3-L1 preadipocytes differentiation or other similar *in vitro* systems. Additional experiments were conducted to explain the discrepancy between the inhibitory effects of EH-CS on adipogenesis gene expression and lipid accumulation.

The inconsistency in inhibition of COX-2 protein by different EH-CS ratios in high dose is critical point and implicating a lot in this study. The most prominent target predicted by compound-target network analysis was PTGS2 (COX-2), which was expected to be modulated by EH and CS (Fig 6). We verified the effects of EH-CS on PTGS2, IL-6, and TNF-a using a well-established PA-induced preadipocyte inflammatory model [54]. Furthermore, the gene expressions of IL-6 and COX-2 were significantly more increased by s4 and s5 at 25 μg/ml than at

10 μg/ml even the concentration was within the limit of cell viability (90% cell viability) (Fig 10A and 10D). In other samples, however, the pro-inflammatory property was not observed. Rather, they showed better anti-inflammatory activity in higher dose. This indicates detrimental effects of EH is successfully controlled by CS while preserving the anti-adipogenic efficacy, allowing us to use the combination safely in higher dose. This phenomenon is not common in other herbal combinations.

As was described, COX-2 mRNA and protein expressions were inhibited by EH-CS combinations, and s3 had the greatest effect (Fig 10A and 10B). It has been reported that modulation of inflammatory status in adipose tissue microenvironment is critical for adipogenesis, thus it is considered a therapeutic target [55]. Interestingly, CS extract (s1) tended to be less effective than EH extract (s5) at modulating the expressions of adipogenic genes or lipid synthesis markers in mature adipocytes (Figs 8 and 9). This suggests compounds in CS act in concert with those in EH to augment the pharmacological effect of EH but suppress EH-induced inflammation. Thus, our results suggest that EH and CS combinations can be used to reduce the amount of EH administered while preserving treatment effectiveness and improving safety.

It has been reported that the extracted amount of ephedrine (a major biologically active constituent of Ephedrae herba) from hot water extract of Ephedrae herba can be reduced by 57–83% when it was combined with other herbs [14]. If we review several obesity-related studies on herbal prescriptions containing the EH-CS combination, the weight ratios of EH to CS used were 40:60 [22], 30:70 [56], or 25:75 [57] or contained an even higher percentage of CS [58], which we ascribe to empirical learning of traditional medicine. Therefore, it appears additional study is needed to fully optimize the EH to CS ratio as per their clinical demands.

As a phytosterol found in many soybeans, stigmasterol has been extensively reviewed for its various benefits on health, including its outstanding anti-oxidant and anti-diabetic activities [59]. It has been reported to attenuate insulin resistance and hyperlipidemia *in vivo*, which are significant clinical features of obesity [60]. Linolenic acid, an essential fatty acid and also a polyunsaturated fatty acid with significant effects on obesity, is the source of the ethyl ester mandenol. Alpha linolenic acid has been reported to improve cholesterol homeostasis in HFD-fed mice model [61], and obesity-associated non-alcoholic liver disease [62]. As we illustrated in network pharmacologic analysis (Fig 6), these compounds might work in combination with other active compounds to attenuate obesity via modulating major targets including PTGS2, ADRB2, and NCOA2.

The present study has several limitations that warrant consideration. First, it is possible that other targets were not included in the network pharmacologic analysis due to their molecular features since we adopted cutoff in OB and DL. Many natural ingredients are known to be metabolized into smaller, biologically active, absorbable molecules by gut microbiota [63] and digestive enzymes [64,65]. Therefore, we suggest that detailed consideration of herbal constituents is required prior to in silico analysis to improve the accuracy and reliability of the data obtained.

Second, we performed *in silico* analysis combined with *in vitro* study, but no *in vivo* study. Thus our results do not take into account several factors related to drug efficacy and side effects associated with digestion, appetite (food intake), the nervous system (especially the potential sympathomimetic effect of EH and its alkaloid ephedrine [66]), and hormonal changes. We carefully estimate that the inhibitory efficacy of the sample on lipid accumulation which is seemingly not outstanding is due to the limitations of the *in vitro* system, where various factors are restricted.

Third, experiments performed at the molecular level were less than comprehensive. Based on compound-target network analysis, common targets of EH and CS, such as PTGS2,

NCOA1, and ADRB2, and their associated molecules were investigated (Fig 6). In the future, we intend to investigate key compounds pair from two herbs. Notably, synergism between caffeine and ephedrine in the context of metabolism is well recognized [67]. However, little information is available on compounds that antagonize or modulate the side effects of ephedrine.

The present study proposes presence of unique combinational effect of two herbs used in treating obesity. Nonetheless, additional *in vivo* and clinical studies are required to confirm that the effects observed in adipocyte model manifest as anti-obesity effects *in vivo*.

## Supporting information

**S1 Graphical abstract.**
(PNG)

**S1 Raw images.**
(PDF)

## Author Contributions

**Conceptualization:** Dong-Woo Lim.

**Funding acquisition:** Won-Hwan Park.

**Methodology:** Dong-Woo Lim, Ga-Ram Yu.

**Supervision:** Won-Hwan Park.

**Validation:** Jai-Eun Kim.

**Writing – original draft:** Dong-Woo Lim.

**Writing – review & editing:** Ga-Ram Yu, Jai-Eun Kim.

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
