## [Decision Letter · Decision Letter 0]

30 Jan 2023

PONE-D-22-34202Network pharmacology predicts combinational effect of novel herbal pair consist of Ephedrae herba and Coicis semen on adipogenesis in 3T3-L1 cellsPLOS ONE

Dear Dr. Kim,

Thank you for submitting your manuscript to PLOS ONE. After careful consideration, we feel that it has merit but does not fully meet PLOS ONE’s publication criteria as it currently stands. Therefore, we invite you to submit a revised version of the manuscript that addresses the points raised during the review process.

We look forward to receiving your revised manuscript.

Kind regards,

Chun-Hua Wang

Academic Editor

PLOS ONE

Journal Requirements:

2. Please ensure all the website links in the manuscript works. For instance, we are unable to access https://old.tcmsp-e.com/tcmsp.php.

 "LDW. (only author that was awarded fund)

This work was supported by the Basic Science Research Program of the Korean National Research Foundation funded by the Ministry of Education (Grant no. 2021R1A6A3A01086718 ).

https://www.nrf.re.kr/index

NO."

Reviewers' comments:

Reviewer's Responses to Questions

**Comments to the Author**

1. Is the manuscript technically sound, and do the data support the conclusions?

Reviewer #1: No

Reviewer #2: Partly

Reviewer #3: Partly

Reviewer #4: Yes

2. Has the statistical analysis been performed appropriately and rigorously? 

Reviewer #1: Yes

Reviewer #2: Yes

Reviewer #3: Yes

Reviewer #4: Yes

3. Have the authors made all data underlying the findings in their manuscript fully available?

Reviewer #1: Yes

Reviewer #2: Yes

Reviewer #3: Yes

Reviewer #4: Yes

4. Is the manuscript presented in an intelligible fashion and written in standard English?

Reviewer #1: No

Reviewer #2: Yes

Reviewer #3: No

Reviewer #4: Yes

5. Review Comments to the Author

Reviewer #1: The present manuscript from Lim and colleagues, entitled “Network pharmacology predicts combinational effect of novel herbal pair consist of Ephedrae herba and Coicis semen on adipogenesis in 3T3-L1 cells” applied network pharmacology analysis to propose a combinational effect of EH-CS herbal pair in treating obesity, and they further evaluated this effect in vitro with a 3T3-L1 adipocyte model.

1. Overall, this study is too simple. The anti-obesity effect of EH-CS combination remains on the surface, some description in the results is inconsistent with the actual figure, and discussion regards distinct effects caused by different doses in the same combination is lacking.

2. The manuscript needs to be polished the English grammar or spelling by a native speaker.

Obvious mistakes are everywhere:

1) Article (the/a) usage problems throughout the manuscript.

2) Abstract: line 27-“ we investigated the compound-target interactions of EH and CS in combination to predict their effects in combination” repeated in combination, the second one is excess.

3) Introduction: the last line is missing punctuation (full stop).

3. As a general rule for writing, titles/labels/legends for tables are always placed at the top of the table, while all the table captions were placed below the tables in this manuscript.

4. The picture does not convey the meaning.

1) In figure 3b, the descriptions don’t correspond to the actual Venn diagram. There is an overlap between CS target and the Obesity target, while the description and the number in the diagram indicate 0.

2) There is no symbol to indicate a significant difference in figure 7a .

3) Figure 7a, 30 μg/ml seems to increase the cell viability instead of decrease it. And I think you mean it reduces the cell viability to 90% instead of reducing by 90%.

5. The quality of the Immunoblot images is poor in Figure 8 and 10, especially the gray analysis results of Western blot are inconsistent with the actual images in figure 10. The original image of the entire PVDF membrane needs to be provided. The internal reference antibody is needed in Western blot in figure 8.

6. Regarding IL6, TNF, and PTGS2 (COX-2) gene changes in palmitate-stimulated (0.5 mM) preadipocytes incubated with the five EH-CS samples, 10 and 25 μg/ml seem to have the opposite effect. But the authors haven’t discussed this inconsistency at all.

Reviewer #2: As the author mentioned in the discussion, CS increased the adiposity of matured 3T3L1; however, lipid contents in the 3T3L1 cells were reduced by CS-EH combined extracts (Fig. 7b).

However, the expression of differentiation-related genes or the degree of phosphorylation of AMPK was not different from that of the EH-only treatment group, suggesting that the mixed treatment of CS and EH would have a substantial lipid reduction effect through the activity of pathways other than the differentiation-related genes. The author then suggested the inhibition of COX-2 as a significant pathway. However, the secretion of inflammatory cytokines such as IL-6 and TNF was not excellent in inhibition compared to the low-concentration EH-only treatment group.

I'd like to hear the author's opinion on this comment.

Reviewer #3: The authors aimed to investigate the combinational effect of novel herbal pair consist of Ephedrae herba and Coicis semen on adipogenesis in 3T3-L1 cells by the Network pharmacology. Although interesting, authors should add and clarify the following requirements:

1. Study title is “Network pharmacology predicts combinational effect of novel herbal pair consist of Ephedrae herba and Coicis semen on adipogenesis in 3T3-L1 cells”. The relationship between the study subjects should be elaborated in detail in the Introduction section.

2. Through network pharmacology analysis, the authors screened three common targets (PTGS2, NCOA1, and ADRB2), two common compounds (stigmasterol and mandenol), and one core target (IL-6) in EH-CS drug pairs. The authors should discuss the effects of these targets and compounds in obesity, separately.

3. In the introduction, the authors describe the beneficial effects of Ephedrae herba and Coicis semen on ameliorating obesity, but do not address why obesity was chosen as a disease to be explored in this study.

4. The authors screened the key targets of EH-CS from the PPI network. However, the authors should describe in more detail how to build the PPI network from the STRING database. Which specific parameters were set and why the parameter 0.4 was chosen?

5. In section 3.4, “The 152 obesity-related target genes were uploaded into the STRING database to obtain a PPI network, which was then analyzed in Cytoscape”. The figure of Cytoscape should be supplemented.

6. Kindly provide the citation details for the methods adopted in Cell Viability Assessment, Oil Red O Staining, and The Palmitate-induced Preadipocyte Inflammatory Model.

7. Result: the figure between 3.7 Cytotoxicity of EH-CS Samples on Preadipocytes and 3.8 Inhibitory Effects of EH-CS on Adipocyte Differentiation should be kept separate for data processing.

Some minor issues:

1. In line 327, is it "Fig.4" or "Fig.5"?

2. The title of the table should be located at the top of the table.

3 It is recommended that authors standardize the expression of numbers, when to express them in English (such as three) and when to express them in Arabic numerals (such as 152)?

4. The full names of TCMSP, OB, and DL in lines 126-127 should be supplemented, while the full names in lines 283-285 should be deleted.

5. Line 376: AMPK phosphorylation ratio should be corrected to AMPK phosphorylation / AMPK?

6. Line 208-209: the revised content should be deleted.

7. This manuscript needs to be checked for spelling and grammar errors

Reviewer #4: Please, check manuscript about spelling check. For example, in line 179, 'CO2' be better to be changed 'CO2(the underscript style of 2)'.

In the addition,

In line 199, '3: 2.' be checked.

In line 246, 'means±SDs' be better to be changed to 'mean ± SDs' (taking a blank).

In line 252, 'com-pounds' be better to be changed to 'compounds'.

In line 247 and 319, 'p-values' be better to be changed to 'P-values (an italic style of 'P').

In line 397, 403, 422, 423, 'P<' or 'P <' needs to be unified into one style.

6. PLOS authors have the option to publish the peer review history of their article (what does this mean?). If published, this will include your full peer review and any attached files.

Reviewer #1: **Yes: **Yuanyuan Deng

Reviewer #2: No

Reviewer #3: No

Reviewer #4: No

---

## [Author Response · Author response to Decision Letter 0]

20 Feb 2023

Notifications to reviewers 

- Annotated Line number in this “Response to reviewers” file indicates Line number in “Revised manuscript with track changes” file. 

- Please also read “Response to reviewers.docx” file for revision figures used in this rebuttal document. Thank you very much. 

Review Comments to the Author 

Reviewer #1: The present manuscript from Lim and colleagues, entitled “Network pharmacology predicts combinational effect of novel herbal pair consist of Ephedrae herba and Coicis semen on adipogenesis in 3T3-L1 cells” applied network pharmacology analysis to propose a combinational effect of EH-CS herbal pair in treating obesity, and they further evaluated this effect in vitro with a 3T3-L1 adipocyte model.

We really appreciate for your sincere comments on the manuscript. Reading your critical opinion greatly helped us to reconsider on publishing our manuscript with the presented data. We feel grateful that we still have a chance to explain the major points of the manuscript to reviewers. 

Many anti-obesity drugs were efficacious in reducing body weight for a short period, but the drug’s effectiveness was not sustained in long-term observation or serious side effects appeared.

Ephedrae herba (EH) has long been used to treat obesity, however, the side effects of EH has been a significant issue by medical practitioners. EH was used in combination with other herbs to treat obesity; many of the cases were CS. We described it in Line 79.

First, we tried to find reasonable evidences for existence of the herbal combination (EH-CS). After reviewing numerous Korean articles on various herbal prescriptions used on obesity, we have concluded that the combination is found frequently and is a significant pair in many prescriptions when compared with other herbal combinations. 

Therefore, we analyzed the interactions between herbs consisting anti-obesity prescription and presented it as 1-mode herb network (Figure 1).

Then we used network pharmacological approaches (NP) to assume the potential mechanisms of the herbal combination. Prior to investigation, principal investigator of this study had the impression of definite roles of EH and CS in prescriptions, each was explained as Monarch (jun, 君) and Minister (chen, 臣) of herbal formula theory in Oriental medicine. 

To our surprise, the idea seems to be sound as demonstrated by NP results. All biological targets of CS were all included in EH targets which can be tentatively interpreted as below;

- The spectrum of whole pharmacological activities of EH is broader than CS.

- CS is supporting or modulating the partial function of EH by targeting the same proteins. 

EH and CS have several targets in common (20 targets) against obesity. These co-regulated targets might explain the decreased toxicity of EH and increased efficacy of EH-CS combination in clinic. 

Cell based study showed combinational effect of the EH-CS. However, the therapeutic efficacy of EH-CS combinations (s1-s5) on lipid accumulation was not corresponding to the anti-obesity mechanism (lipogenesis or adipose differentiation). But we found significant benefit of using EH-CS combination in suppressing deleterious effects induced by EH by inhibiting inflammatory markers. 

We have addressed all issues commented by reviewers. We really hope reviewers are satisfied with our revised manuscript. 

1. Overall, this study is too simple. The anti-obesity effect of EH-CS combination remains on the surface, some description in the results is inconsistent with the actual figure, and discussion regards distinct effects caused by different doses in the same combination is lacking.

We feel sorry if you felt the data of our manuscript is too simple. 

However, although it may seem simple, this paper has done three dimensions of analysis; Literature review, network pharmacologic analysis, and in vitro analysis. Also these data are supported by HPLC analysis.

We admit that in vitro experiment data is not rigorous; however, we insist that the in vitro data is not the only part in this paper.

We tried to improve the quality of this manuscript by responding all of comments by reviewers. 

The inconsistency of efficacy on inhibiting pro-inflammatory markers by s4 and s5 between two doses (10 and 25 μg/ml) implicates a lot in this study. The issue is addressed in reviewer comments 6.

(6. Regarding IL6, TNF, and PTGS2 (COX-2) gene changes in palmitate-stimulated (0.5 mM) preadipocytes incubated with the five EH-CS samples, 10 and 25 μg/ml seem to have the opposite effect. But the authors haven’t discussed this inconsistency at all.)

We reinforced the discussion part to explain the meaning of inconsistency. .

In discussion part Line 492, we added 

- “The inconsistency in inhibition of COX-2 protein by different EH-CS ratios in high dose is critical point and implicating a lot in this study.” 

In discussion part Line 497, we added 

- “even the concentration is within the limit of cell viability (90% cell viability)”

In discussion part Line 498, we added 

- “In other samples, however, the pro-inflammatory property was not observed. Rather, they showed better anti-inflammatory activity in higher dose. This indicates detrimental effects of EH was successfully controlled by CS while preserving the anti-adipogenic efficacy, allowing us to use the combination safely in higher dose. This phenomenon is not common in other herbal combinations.”

2. The manuscript needs to be polished the English grammar or spelling by a native speaker.

Obvious mistakes are everywhere:

1) Article (the/a) usage problems throughout the manuscript.

We are terribly sorry for the grammatical errors. Despite we had thorough English proofreading by native speaker, some mistakes are still remaining after minor manuscript revision.

2) Abstract: line 27-“ we investigated the compound-target interactions of EH and CS in combination to predict their effects in combination” repeated in combination, the second one is excess.

We thank you for your comment. We removed the excessive ‘in combination’ from the manuscript.

3) Introduction: the last line is missing punctuation (full stop).

We corrected the sentence by adding punctuation.

3. As a general rule for writing, titles/labels/legends for tables are always placed at the top of the table, while all the table captions were placed below the tables in this manuscript.

We appreciate for your valuable comment on placement of captions.

We admit that it needs to be placed correctly. So we corrected it immediately.

4. The picture does not convey the meaning.

1) In figure 3b, the descriptions don’t correspond to the actual Venn diagram. There is an overlap between CS target and the Obesity target, while the description and the number in the diagram indicate 0.

We are grateful for giving us a chance to explain.

As we described in our manuscript, all CS targets (31 targets) were included in EH targets (described in result chapter 3.3 Active compound screening and key targets of EH-CS in combination; ‘interestingly, the 31 CS targets were included among EH targets.’). So, there is not a single target that only belongs to CS.

That explains the ‘0’ in the venn diagram. We hope you satisfied with our explanation.

2) There is no symbol to indicate a significant difference in figure 7a .

We appreciate for your comment on the significance test result of Figure 7a.

We performed the statistical analysis with cell viability data (One way anova with Tukey post-hoc test, Revision Figure 1). However, there was no significant reduction in cell viability as per the statistical analysis.

However, as we considered 90 percent of cell viability as good condition, we used 25 μg/ml as maximal concentrations of in vitro study. 

In some cases, viability (mean value as percentage) is heavily reduced while the statistical significance is not met (p>0.05). And opposite case is also possible. In our case, we think that the mean value of cell viability should come first than the statistical significance, since we are intended to find safe dose for further in vitro study. 

We hope you are satisfied with our explanation.

Revision Figure 1 Statistical analysis result of cell viability.

3) Figure 7a, 30 μg/ml seems to increase the cell viability instead of decrease it. And I think you mean it reduces the cell viability to 90% instead of reducing by 90%.

We appreciate for your valuable comments.

We confirmed the raw data of cell viability. No error was found in the raw data, but unfortunately, SD value was relatively large in the concentration (30 μg/ml).

Despite we cannot rule out the possibility that it was due to experimental or measurement error, we consider 30 μg/ml of EH seems to be safe for cell condition. 

We needed to choose maximal concentration considering stability of all cells treated with five samples. In case of the s5 sample, the concentration value for cell viability seems to lay around 28.6 μg/ml, which was estimated by linear regression estimation. We described the deduction process in below (Revision Figure 2). 

Therefore, we acknowledged that the concentration of 25 μg/ml was acceptable and continued in vitro study.

To avoid possible misunderstanding, we corrected the word (by 90%) as you recommended (to 90%). 

Revision Figure 2 Estimation of limit concentration of s5 for 90% cell viability

5. The quality of the Immunoblot images is poor in Figure 8 and 10, especially the gray analysis results of Western blot are inconsistent with the actual images in figure 10. The original image of the entire PVDF membrane needs to be provided. The internal reference antibody is needed in Western blot in figure 8.

Thank you for giving us sincere comments on western blot images.

We want to be clear that the results analysis and blot images are accurate. We feel sorry that the standard deviations (SDs) of blot intensity are relatively large in some conditions. We did three times of western blot (the COX-2 protein). We obtained immunoblot images from three independent experiments on different days (We assume that it’s the reason for large SDs).

We already submitted all our western blot images to journal. We also attached the images in here for reviewer to confirm (Review Figure 3). Please check it.

We are not sure what exactly ‘the reference antibody’ means. In case you mean housekeeping gene like beta-actin as ‘the reference antibody’, here we presented with beta-actin images for the phosphorylated-AMPK/AMPK signals (Review Figure 4). 

We also revised figure 8 by adding beta-actin images. We really hope you are satisfied with our explanation.

Revision Figure 3 Three immunoblot images of COX-2 obtained in different days (for Figure 10b).

Revision Figure 4 Immunoblot images of beta-actin obtained in different days (for Figure 8a)

6. Regarding IL6, TNF, and PTGS2 (COX-2) gene changes in palmitate-stimulated (0.5 mM) preadipocytes incubated with the five EH-CS samples, 10 and 25 μg/ml seem to have the opposite effect. But the authors haven’t discussed this inconsistency at all.

We would like to thank you for bringing up critical points on discussion part.

As you pointed out, the inconsistency of COX-2 protein inhibition by different EH-CS ratios is critical point and implicating a lot in this study. 

As we presented, the cell viability was notably affected by EH treatment. Moreover, the inflammation was induced at 25 μg/ml of s4 and s5 recognized by increased COX-2 protein and IL6 mRNA levels.

The point is that, when cells were treated with EH-enriched samples (s4, s5) at 25 μg/ml, the samples have detrimental effects even though the concentration is within the limit concentration (90% cell viability). This phenomenon is not usual with other herbal samples.

However, in other samples, the inflammation-inducing property was not observed at high concentration. Rather, they showed better activity (inflammatory inhibition) in higher concentration. This might be due to the pharmacological activities of CS and is what we wanted to emphasize in our study.

Our team is already conducting next study using this EH-CS combination to investigate the synergistic efficacy using combination of single compound (from CS) and herb (EH).

We added some points in discussion part to emphasize the significance of our findings.

- In Line 500, “This indicates detrimental effects of EH is successfully controlled by CS while preserving the anti-adipogenic efficacy, allowing us to use the combination safely in higher dose. This phenomenon is not common in other herbal combinations.”

Reviewer #2: As the author mentioned in the discussion, CS increased the adiposity of matured 3T3L1; however, lipid contents in the 3T3L1 cells were reduced by CS-EH combined extracts (Fig. 7b).

However, the expression of differentiation-related genes or the degree of phosphorylation of AMPK was not different from that of the EH-only treatment group, suggesting that the mixed treatment of CS and EH would have a substantial lipid reduction effect through the activity of pathways other than the differentiation-related genes. 

The author then suggested the inhibition of COX-2 as a significant pathway. However, the secretion of inflammatory cytokines such as IL-6 and TNF was not excellent in inhibition compared to the low-concentration EH-only treatment group.

I'd like to hear the author's opinion on this comment.

We appreciate for your sincere comments on our manuscript. We are grateful that we had another chance to explain about it.

We humbly suggest you to focus on the other area than the AMPK and its downstream pathways. 

In fact, this study was intended to find the reason why combination is necessarily used rather than the EH or CS alone. As you can see, the EH-CS was not showing outstanding efficacy in reducing lipid accumulation in matured adipocytes, as compared to other prominent materials used in other studies by peer researchers.

In our opinion, the reason is the efficacy of CS in controlling side effects caused by EH. Therefore, we predicted the common targets to be the mechanisms to control side effects. We further investigated the network pharmacology-deduced common targets of CS and EH. 

As you can see in figure 8 and 9, EH alone exerted strongest inhibiting activity in adipogenesis along with enhanced activity of AMPK. However, ironically, the inhibition of lipid accumulation in s5 was poor. This needed to be explained. 

We investigated the expression of COX-2, which is a remarkable inflammatory marker and common target of both EH and CS. It showed significant increase in high dose (25 μg/ml) of EH-enriched samples (s4 and s5) as compared to low dose (10 μg/ml), even the concentration is within the limit concentration of cell viability. This is not usual phenomenon in other drugs. We suspect this phenomenon was induced due to detrimental effects of EH on cell physiology with unidentified mechanism. It is well known that the pro-inflammatory condition can significantly stimulate adipogenesis in preadipocytes.

However, the s3 or other CS-enriched samples (s1, s2) showed better performances in reducing inflammatory markers expression in higher dose (except few cases). This means that, EH can be used safer and better when combined with CS. 

In summary, EH has strong efficacy in inhibiting adipogenesis while it also has inflammatory/toxicity issue. But the use of EH in combination with CS greatly resolves the issue while it preserving the efficacy. 

This type of combinational effect has not been reported very much. This mutual pharmacological activity of two herbs is somewhat similar to what they described in herbal formula theory in Oriental medicine. 

The usage of two herbs in combination has been in empirical field used by practitioners. We think now is the right time to report the herbal combination into scientific field.

Reviewer #3: The authors aimed to investigate the combinational effect of novel herbal pair consist of Ephedrae herba and Coicis semen on adipogenesis in 3T3-L1 cells by the Network pharmacology. Although interesting, authors should add and clarify the following requirements:

1. Study title is “Network pharmacology predicts combinational effect of novel herbal pair consist of Ephedrae herba and Coicis semen on adipogenesis in 3T3-L1 cells”. The relationship between the study subjects should be elaborated in detail in the Introduction section.

We are grateful for your sincere comments on introduction part. As you pointed out, the rationale for using network pharmacologic approach needs to be described in introduction.

Therefore, in Line 86-88, we wrote as “Network pharmacology, an integrative tool for analyzing pharmacological mechanism of herbal medicine [23], can be used to decipher complex interactions between numerous targets and compounds derived from EH and CS.” 

2. Through network pharmacology analysis, the authors screened three common targets (PTGS2, NCOA1, and ADRB2), two common compounds (stigmasterol and mandenol), and one core target (IL-6) in EH-CS drug pairs. The authors should discuss the effects of these targets and compounds in obesity, separately.

We appreciate your sincere advice to write more about important targets and compounds discovered through network pharmacology.

We wrote more about key targets in line 89-98 with new references ([24-28]).

“Prostaglandin-endoperoxide synthase2 (PTGS2) is inducible enzyme expressed in inflamed conditions leading to biosynthesis of prostaglandins [24]. It has been noted that inhibition of COX activity affects adipocyte differentiation via decreasing inflammatory cytokines [25]. A member of the nuclear hormone receptor coactivator family, nuclear receptor coactivator 2 (NCOA2) controls adipogenesis, lipid metabolism, and fat absorption to maintain metabolic balance [26]. Adrenoreceptor beta 2 (ADRB2) is connected to elevated noradrenaline release brought on by exposure to cold, which activates lipolysis and thermogenesis [27]. Interleukin-6 (IL6) is pro-inflammatory mediator which is suggested as cause of systemic low grade inflammation in obesity [28]. By offering a variety of strategies in various metabolic pathways, those genes can be used as targets for the treatment of obesity.” 

We also wrote more about key compounds in line 519-527 with new references ([59-62])

“As a phytosterol found in many soybeans, stigmasterol has been extensively reviewed for its various benefits on health, including its outstanding anti-oxidant and anti-diabetic activities [59]. It has been reported to attenuate insulin resistance and hyperlipidemia in vivo, which are significant clinical features of obesity [60]. Linolenic acid, an essential fatty acid and also a polyunsaturated fatty acid with significant effects on obesity, is the source of the ethyl ester mandenol. Alpha linolenic acid has been reported to improve cholesterol homeostasis in HFD-fed mice model [61], and obesity-associated non-alcoholic liver disease [62]. As we illustrated in network pharmacologic analysis (Fig 6), these compounds might work in combination with other active compounds to attenuate obesity via modulating major targets including PTGS2, ADRB2, and NCOA2.”

3. In the introduction, the authors describe the beneficial effects of Ephedrae herba and Coicis semen on ameliorating obesity, but do not address why obesity was chosen as a disease to be explored in this study.

Thank you for giving us kind comments on introduction part.

As you indicated, EH-CS combination has been frequently used in treating obesity by clinical practitioners. This was clearly described in introduction part (Line 79-82).

We think it is a basic strategy to investigate efficacy of herbs (or natural products) as per their traditional medicinal usages. 

4. The authors screened the key targets of EH-CS from the PPI network. However, the authors should describe in more detail how to build the PPI network from the STRING database. Which specific parameters were set and why the parameter 0.4 was chosen?

Thank you for raising an important point on construction of PPI network.

The parameter of 0.4 was chosen as it is a figure of medium confidence which is most frequently used. If we raise the criteria, the interactions between target proteins will be loosening as a result.

There was no other modification in setting up PPI network. In fact, there aren’t many modifiable settings in STRING, except some options related to visualization (like font, presentation of protein name, disable bubble design etc.). These options does not affect the structure of network .

5. In section 3.4, “The 152 obesity-related target genes were uploaded into the STRING database to obtain a PPI network, which was then analyzed in Cytoscape”. The figure of Cytoscape should be supplemented.

We welcome your valuable suggestion.

We think that the total PPI network consists of 152 obesity-related target is not much informative for readers. Including it in the manuscript would consume too much space, as the figure does not provide any crucial information. The total network was only used for extracting core network in this study.

Furthermore, the core network consists of 31 target genes from total network is already attached as figure 4.

Therefore, we decided not to insert the image in main manuscript. If you insist, however, we will change it. 

Here we attached the original image of 152 obesity-related PPI network for your understanding (Revision figure 5). We are confident that reviewer will agree with our opinion.

Revision Figure 5 Total PPI network of EH-CS consists of 152 target genes. 

6. Kindly provide the citation details for the methods adopted in Cell Viability Assessment, Oil Red O Staining, and The Palmitate-induced Preadipocyte Inflammatory Model.

We feel gratitude for suggesting us good points on references for specific experimental methods.

We added two references ([41] for cell viability assessment, [42] for oil red o staining) as per your suggestion.

Reference [43] is already cited in the main manuscript for palmitate-induced inflammatory model for preadipocytes.

7. Result: the figure between 3.7 Cytotoxicity of EH-CS Samples on Preadipocytes and 3.8 Inhibitory Effects of EH-CS on Adipocyte Differentiation should be kept separate for data processing.

We appreciate your valuable comments on figure 7.

We understand that you asked the two figures (Fig. 7A and Fig. 7B) should be separated into different figures (Fig. 7, Fig. 8). 

However, we assume this rearrangement of figure will deteriorate overall quality of the manuscript and give bad impression since the total number of figures is going to be large (total 11 figures).

We gently ask you to withdraw the opinion. However, if you insist, then we will separate the figure as you requested. 

Some minor issues:

1. In line 327, is it "Fig.4" or "Fig.5"?

We confirmed it. The Fig. 4 is correct. The figure 4 is annotated in both section 3.4 and 3.5. 

2. The title of the table should be located at the top of the table.

We checked and corrected it immediately.

3 It is recommended that authors standardize the expression of numbers, when to express them in English (such as three) and when to express them in Arabic numerals (such as 152)?

We regarded it as reasonable to express numbers in English when it modifies a significant numbers of sample, trials, core targets less than 10. On the contrary, huge numbers with less significance were expressed in Arabic numerals as it doesn’t need to be expressed in English.

If you still think it needs to be changed, we will gladly change it.

4. The full names of TCMSP, OB, and DL in lines 126-127 should be supplemented, while the full names in lines 283-285 should be deleted.

Sorry for the mistake. We corrected it.

5. Line 376: AMPK phosphorylation ratio should be corrected to AMPK phosphorylation / AMPK?

We agree with your suggestion. We corrected it.

6. Line 208-209: the revised content should be deleted.

We corrected it by using track change function. 

7. This manuscript needs to be checked for spelling and grammar errors

We fully admit that our manuscript needs to be checked for grammatical errors.

Other reviewers also pointed out the same issue. We read the manuscript carefully and revised it. 

We appreciate again for your warm and kind comments. 

Reviewer #4: Please, check manuscript about spelling check. For example, in line 179, 'CO2' be better to be changed 'CO2(the underscript style of 2)'.

We appreciate for your kind comments on our manuscript. We carefully read and improved the quality of our manuscript.

We replaced it with underscript character (CO2). 

In the addition,

In line 199, '3: 2.' be checked.

We checked it. We changed it to '3 : 2’ (spacing after 3). 

In line 246, 'means±SDs' be better to be changed to 'mean ± SDs' (taking a blank).

We corrected it as per your suggestion.

In line 252, 'com-pounds' be better to be changed to 'compounds'.

Sorry for the mistake. It must be derived from the copy-pasting process from the last version of manuscript. We corrected it immediately. 

In line 247 and 319, 'p-values' be better to be changed to 'P-values (an italic style of 'P').

We agree with your suggestion on Italicizing ‘P’ used in ‘P-value’. We corrected it in line 247 and 319. (Line 268 and 343 in revised manuscript)

In addition, we Italicized P in Line 179, 180 (Line in revised manuscript). Also, we changed P to Italics in Table 4.

In line 397, 403, 422, 423, ‘P<’ or ‘P <’ needs to be unified into one style.

We decided to unify all space rule after ‘less than sign’ (<).

Therefore, we corrected with it throughout the manuscript to have space before ‘<’ sign.

We want to express our deepest thank for your kind comments.

---

## [Editor Report · Decision Letter 1]

27 Feb 2023

Network pharmacology predicts combinational effect of novel herbal pair consist of Ephedrae herba and Coicis semen on adipogenesis in 3T3-L1 cells

PONE-D-22-34202R1

Dear Dr. Kim,

We’re pleased to inform you that your manuscript has been judged scientifically suitable for publication and will be formally accepted for publication once it meets all outstanding technical requirements.

Kind regards,

Chun-Hua Wang

Academic Editor

PLOS ONE

Additional Editor Comments:

The authors have addressed the issues raised by the reviewers, so, it can be accepted in this version.

---

## [Editor Report · Acceptance letter]

2 Mar 2023

PONE-D-22-34202R1 

Network pharmacology predicts combinational effect of novel herbal pair consist of Ephedrae herba and Coicis semen on adipogenesis in 3T3-L1 cells 

Dear Dr. Kim:

I'm pleased to inform you that your manuscript has been deemed suitable for publication in PLOS ONE. Congratulations! Your manuscript is now with our production department. 

Kind regards, 

on behalf of

Dr. Chun-Hua Wang 

Academic Editor

PLOS ONE